# *APC* mutations in human colon lead to decreased neuroendocrine maturation of ALDH+ stem cells that alters GLP-2 and SST feedback signaling: Clue to a link between WNT and retinoic acid signalling in colon cancer development

Tao Zhang[1,2,3‡], Koree Ahn[1,2,3‡], Brooks Emerick[4], Shirin R. Modarai[1,2], Lynn M. Opdenaker[1,2], Juan Palazzo[3], Gilberto Schleiniger[4], Jeremy Z. Fields[5], Bruce M. Boman[1,2,3,4]*

**1** Cawley Center for Translational Cancer Research, Helen F. Graham Cancer Center and Research Institute, Newark, DE, United States of America, **2** University of Delaware, Newark, DE, United States of America, **3** Thomas Jefferson University, Philadelphia, PA, United States of America, **4** Center for Applications of Mathematics in Medicine, Department of Mathematical Sciences, University of Delaware, Newark, DE, United States of America, **5** CATX, Inc., Princeton, NJ, United States of America

‡ Co-first authors who contributed equally to the study.
* brboman@udel.edu

## Abstract

*APC* mutations drive human colorectal cancer (CRC) development. A major contributing factor is colonic stem cell (SC) overpopulation. But, the mechanism has not been fully identified. A possible mechanism is the dysregulation of neuroendocrine cell (NEC) maturation by *APC* mutations because SCs and NECs both reside together in the colonic crypt SC niche where SCs mature into NECs. So, we hypothesized that sequential inactivation of *APC* alleles in human colonic crypts leads to progressively delayed maturation of SCs into NECs and overpopulation of SCs. Accordingly, we used quantitative immunohistochemical mapping to measure indices and proportions of SCs and NECs in human colon tissues (normal, adenomatous, malignant), which have different *APC*-zygosity states. In normal crypts, many cells staining for the colonic SC marker ALDH1 co-stained for chromogranin-A (CGA) and other NEC markers. In contrast, in *APC*-mutant tissues from familial adenomatous polyposis (FAP) patients, the proportion of ALDH+ SCs progressively increased while NECs markedly decreased. To explain how these cell populations change in FAP tissues, we used mathematical modelling to identify kinetic mechanisms. Computational analyses indicated that *APC* mutations lead to: 1) decreased maturation of ALDH+ SCs into progenitor NECs (not progenitor NECs into mature NECs); 2) diminished feedback signaling by mature NECs. Biological experiments using human CRC cell lines to test model predictions showed that mature GLP-2R+ and SSTR1+ NECs produce, via their signaling peptides, opposing effects on rates of NEC maturation via feedback regulation of progenitor NECs. However, decrease in this feedback signaling wouldn't explain the delayed maturation because both

**Data Availability Statement:** The tissue model is described in the appendix (Supplemental material). The code for the tissue model is available on request.

**Funding:** This study was supported in part by The Lisa Dean Moseley Foundation (BB), NIH Grant 5R21DK62146 (BB), University of Delaware awards for a graduate fellowship stipend from the Department of Mathematical Sciences (BE) and Department of Biological Sciences Graduate Scholars Award (SM)., a Hamline University Summer Student Fellowship Award (KA), Cancer B*Ware Foundation funding (BB, SM, LO), and Cawley Center for Translational Cancer Research Funds (BB, SM, LO). Additional support for the Bioimaging Core Facility was made possible through funding from Delaware INBRE NIH/NIGMS GM103446 (BB, SM, LO). No funding was provided by CATX, Inc. CATX Inc did not provide any support in the form of salaries for Dr Fields or other authors, or provide any other financial backing to the study. Dr Fields has no ownership in CATX, Inc. CATX Inc holds no intellectual property or any commercial interests in the study.

**Competing interests:** The authors have declared that no competing interests exist. CATX Inc is a commercial entity, but no competing conflicts of interest exist including any other relevant declarations relating to employment, consultancy, patents, products in development, or marketed products, etc. CATX Inc does not alter adherence to all PLOS ONE policies on sharing data and materials as specified in the following statement: "This does not alter our adherence to PLOS ONE policies on sharing data and materials."

**Abbreviations:** SC, stem cell(s); CSC, cancer stem cell(s); NE, neuroendocrine; PNC, progenitor neuroendocrine cell; NEC, mature neuroendocrine cell(s); ALDH, aldehyde dehydrogenase; CGA, chromogranin-A; IHC, immunohistochemistry; IF, immunofluorescence; GLP-2, Glucagon-like peptide 2; SST, Somatostatin; SSTR1, Somatostatin receptor 1; GLP-2R, Glucagon-like peptide 2 receptor; RA, retinoic acid; WNT, wingless-int.

progenitor and mature NECs are depleted in CRCs. So the mechanism for delayed maturation must explain how *APC* mutation causes the ALDH+ SCs to remain immature. Given that ALDH is a key component of the retinoic acid (RA) signaling pathway, that other components of the RA pathway are selectively expressed in ALDH+ SCs, and that exogenous RA ligands can induce ALDH+ cancer SCs to mature into NECs, RA signaling must be attenuated in ALDH+ SCs in CRC. Thus, attenuation of RA signaling explains why ALDH+ SCs remain immature in *APC* mutant tissues. Since *APC* mutation causes increased WNT signaling in FAP and we found that sequential inactivation of *APC* in FAP patient tissues leads to progressively delayed maturation of colonic ALDH+ SCs, the hypothesis is developed that human CRC evolves due to an imbalance between WNT and RA signaling.

## Introduction

Our goal was to determine how mutations in *APC* drive colorectal cancer (CRC) development in humans by causing colonic stem cell (SC) overpopulation. To investigate this mechanism, we used ALDH1 as a marker for normal and malignant human colonic SCs. Specifically, we used ALDH1 to track increases in SC population size in colonic crypts from familial adenomatous polyposis (FAP) patients. We chose FAP because it is an ideal human model for hereditary CRC development due to *APC* mutations. Because SCs and neuroendocrine cells (NECs) both reside together in the SC niche of the colonic crypt, and NECs are known to regulate crypt cell proliferation, we investigated the possibility that dysregulation of NECs by *APC* mutations is key to the SC overpopulation.

### *APC* mutations drive CRC development

Several independent lines of evidence demonstrate that *APC* mutation is the primary driving mechanism in human CRC: (i) WNT signalling is altered in ~95% of human CRCs, primarily due to biallelic mutation of the *APC* gene [1]. (ii) *APC* mutation alone is sufficient for early CRC development [2]. (iii) Those CRCs that develop because of an *APC* mutation are associated with a worse prognosis than those CRCs that develop because of DNA mismatch repair mutations [3]. (iv) The extent of *APC* mutation (most mutations lead to truncation of the protein product) correlates with the severity of the tumor [4]. (v) The characteristics of the second *APC* hit depend on the nature of the 1st *APC* hit in the two hit mechanism for CRC [5, 6]. (vi) *APC* mutations are required for the maintenance of colon carcinomas [7]. (vii) Transfection of *APC* into CRC cells induces cell cycle arrest and apoptosis [8, 9]. (viii) Restoring wild-type *Apc* expression in CRCs leads to cellular differentiation and re-establishes crypt homeostasis [10]. (ix) *APC* mutations lead to increased crypt fission, which is the main mechanism in adenoma morphogenesis [11–13].

### FAP is a human genetic model for CRC development due to *APC* mutations

To investigate the mechanisms underlying the ability of *APC* mutations to drive CRC development in humans, we studied tissues from hereditary colon cancer patients from familial adenomatous polyposis (FAP) families. Indeed, investigations of FAP led to the identification, mapping and isolation of the *APC* gene. FAP is an autosomal dominant trait [14] caused by inheritance of a germline *APC* mutation. FAP patients develop 100s to 1000s of premalignant

adenomas which further supports the idea that *APC* mutations drive tumor growth *in vivo*. If FAP patients are left untreated, they have nearly a 100% risk of developing CRC. Adenomas in FAP patients show loss of the 2nd wild type *APC* allele [15–17]. Two hits at the *APC* locus also occur as acquired mutations in the development of most sporadic CRCs. Thus, while FAP is relatively rare (incidence = $1.90 \times 10^{-6}$; prevalence = $4.65 \times 10^{-5}$); [18]), results reported here should have wider implications for understanding mechanisms involved in the development of commonly occurring sporadic adenomatous polyps (1 in 2 individuals) and sporadic CRC (1 in 20 individuals) in the general population [19].

## *APC* mutations lead to SC overpopulation in the colon

Our initial studies [20] of the proliferative abnormality [21] in crypts from FAP patients indicated that an *APC* mutation causes SC overpopulation during initiation of colon tumorigenesis. We validated this finding biologically for pre-malignant and malignant colon tumors from FAP patients. For example, using markers for crypt base cells we found that the crypt base cell population (including SCs) increased in size during adenoma development [22]. In another study [23], we discovered that ALDH1 is a marker for normal and malignant human colonic SCs, and that ALDH1 can be used to track increases in SC population size in colonic crypts from FAP patients during colon tumorigenesis. In a murine model of FAP, *Apc*<sup>Min/+</sup> mice, we found [24] further evidence that *Apc* mutation produces crypt SC overpopulation during intestinal tumorigenesis. Studies by other investigators using different SC markers also suggest that SC overpopulation occurs during intestinal tumorigenesis in humans as well as in mice [25–33]. Further studies [34, 35] indicate that increased symmetric SC division is a mechanism that explains SC overpopulation and drives growth of colonic tumors. These lines of evidence imply that *APC* mutations that drive growth of intestinal tumors do so by causing SC overpopulation [36, 37].

## ALDH is a widely used as a marker for stem cells in humans

We have been using ALDH as a marker to identify and isolate SCs from patient tissues because it is a reliable marker for normal and malignant human colonic SCs in our laboratory [23, 38–41] as well as in other laboratories [42–44]. ALDH is not only a marker for SCs, but also ALDH+ cells have SC properties of self-renewal, cell differentiation potential, and drug resistance. Indeed, the self-renewal property of ALDH+ cells is demonstrated by sphere-forming ability *in vitro* and tumor-initiating ability in mice [23, 43–45]. The cell differentiation property of ALDH+ cells is due to ALDH's functional role as a key enzyme in the retinoic acid (RA) signaling pathway [46–50]. We investigated ALDH+ cells from normal and malignant colonic tissues and found that retinoid receptors RXR and RAR are selectively expressed in ALDH+ cells [39], which indicates that RA signalling mainly occurs via ALDH+ SCs. That RA signalling primarily occurs in ALDH+ cells provides a mechanism for selective treatment of SCs using RA analogues. Indeed, we have shown that treatment of ALDH+ cancer SCs (CSCs) with all-trans retinoic acid (ATRA) inhibits cell proliferation, decreases SC proliferation, sphere formation, and SC population size, as well as increases SC differentiation into NECs [38, 39]. Other investigators also commonly use ATRA as a differentiation agent in SC research [51–54]. And, ATRA is an effective drug used clinically to treat acute promyelocytic leukemia (PML) patients because ATRA can induce PML cells to differentiate into neutrophils [55–57]. The drug resistance property of ALDH+ SCs comes from aldehyde dehydrogenase's enzymatic function, which is the cell's natural detoxification mechanism to protect itself against various aldehydes that would otherwise have harmful consequences [46–50]. These effects are seen with the use of alkylating

agents such as chemotherapeutic agents for treatment of cancers and that ALDH confers resistance of CSCs to cytotoxic agents, particularly the alkylating agent cyclophosphamide [42, 58, 59]. Because a strong body of scientific evidence shows that ALDH has functional properties of SCs, it continues to be used to identify and isolate SCs from colonic tissues as well as from most other human tissue types including breast, lung, prostate, pancreas, hematopoietic, stomach, kidney, brain and others [46, 60–66].

Similarly, ALDH has been used to identify SCs in mouse tissues, including intestine [67. 68], prostate [69], breast [70, 71], hematopoietic [67, 68, 72], and neural [73]. In fact, in the early 1990s, researchers reported that murine intestinal epithelium has high levels of ALDH activity and intestinal crypt SCs displayed high levels of cytosolic ALDH [67, 68]. Still, ALDH was never extensively studied as a marker for mouse intestinal SCs, possibly because there are multiple ALDH isoforms that makes it difficult to do lineage tracing studies. Indeed, there are 19 ALDH isoforms encoded by 19 different genes in humans with as many orthologs in the mouse plus several alternatively-spliced transcriptional variants [46–50, 74, 75]. Consequently, it is not known if ALDH that marks colonic SCs in humans corresponds to any of the many SC markers in murine intestine that have been the focus of intense study in mouse research for many years [76–78]. Indeed, there are many distinct differences between intestine and colon, including differences between mouse colon and human colon [78–83]. However, we have been studying ALDH as a human colonic SC marker (for the reasons cited above) to gain insight into SC-based mechanisms that lead to CRC development in the colon. Perhaps, identifying differences between mechanisms that regulate intestinal SCs versus colonic SCs may help explain why cancer is common in human colon, but rare in small intestine. Indeed, FAP patients who carry *APC* germline mutations develop multiple polyposis and cancers in the colon but few tumors in the small intestine.

## The role of NECs in the SC niche of the colonic crypt

To gain further insight into the role of SCs in CRC development, we investigated NECs because two of the main cell populations in the SC niche in human colonic crypts are SCs and NECs [23, 82–85]. Neuroendocrine cells of the gut are the most abundant endocrine cells in the body [86–88]. And, at least 15 different cell types of neuroendocrine cells are found in the GI tract, which comprise about 1% of the total GI epithelial cell population [89–91]. While the majority of different NEC types are found in small intestine which produce many neurotransmitters, there are just three neuropeptide types produced by neuroendocrine cells in the human colon: glucagon, somatostatin, and bovine pancreatic peptide (BPP), as well as the catecholamine 5-hydroxytryptamine (5-HT). Paneth cells are another specialized secretory cell type found in the SC niche of small intestine but not in colon. But a unique cell type, Reg4 + deep crypt secretory, was recently identified in the colon that is functionally equivalent to Paneth cells that support Lgr5+ SCs [92]. These Reg4+ cells have a NEC phenotype in mouse intestine, but not in the colon. Although there are differences in NEC types between small intestine and colon, in both cases, several lines of evidence indicate that NECs might regulate SCs in the crypt SC niche: (i) NECs secrete factors that can modulate rates of crypt cell proliferation [85, 93–97]; (ii) We previously reported that NEC signaling via the somatostatin pathway contributes to the quiescence of human colon cancer SCs [38]; (iii) In mouse organoid cultures, induced quiescence of LGR5+ SCs by combined blockade of MEK/WNT/NOTCH signaling leads to NEC differentiation [98]; (iv) We also discovered that treatment of human ALDH+ SCs with all-trans retinoic acid (ATRA) induces SC differentiation along the NEC lineage and decreases sphere formation as well as the ALDH+ SC population size [39]. Thus, we surmised that changes occurring in NEC populations during colon tumorigenesis might

lead to dysregulation of crypt SCs which could provide a mechanism that explains how SC overpopulation leads to CRC development.

## Results

To investigate the effect of *APC* mutations on mature NEC, progenitor neuroendocrine cell (PNC), and SC populations during colon tumorigenesis, we addressed the following questions: (i) Do SCs mature along the NE lineage within the SC niche of human colonic crypts? (ii) With sequential mutational inactivation of *APC* in neoplastic crypts, does SC maturation along the NE lineage becomes progressively delayed? Accordingly, we quantified, using immunohistochemical mapping and immunofluorescence co-staining, SCs, PNCs, and NECs in normal, pre-malignant, and malignant human colon tissues, which are known to have different *APC*-zygosity states that represent the stepwise development of CRC. Mathematical modelling was then done to discover kinetic mechanisms that explain how SC, PNC and NEC cell populations change in FAP tissues during CRC development. Finally, we performed *in vitro* biological experiments CRC cell lines to test the predictions from mathematical modelling.

### Quantitative immunohistochemical mapping shows that the proportion of ALDH+ cells progressively increases while CGA+ cells markedly decreases in *APC*-mutant colon tissues from FAP patients

**Normal crypts (homozygous wildtype APC).**   To evaluate the number and distribution of NECs in normal colonic crypts, we first examined the expression of CGA, a marker that labels most types of NECs in most tissues, including colon. Fig 1A and 1B show that cells staining positively for CGA and for ALDH were mostly restricted to the lower crypt. Indeed, staining indices (Fig 1C) show that the crypt distribution of CGA+ cells and ALDH+ cells was similar with the maximum proportion of cells that stained positively for CGA and for ALDH1 occurring at the crypt bottom. The majority of the ALDH1+ and CGA+ cells (~65%) were located in the SC niche (crypt levels 1–20; [99]). The index curve for CGA significantly overlapped the index curve for ALDH1 (Fig 1C). In fact, the two curves were almost superimposable. Calculation of the AUC for the CGA staining index indicated that 5.3% of crypt cells were CGA+. Only slightly more crypt cells, 5.6%, were ALDH+ cells.

**FAP patient tissues (APC mutant crypts).**   To evaluate changes in NEC populations during the stepwise pathologic progression to colon cancer, we examined the above markers (Fig 1A and 1B) in neoplastic colon tissues with varying *APC* genotypes: normal-appearing FAP crypts (heterozygous mutant *APC*); adenomatous FAP crypts (homozygous mutant *APC*); colon carcinomas (homozygous mutant *APC* & other mutations). In normal-appearing FAP crypts, cells staining positively for CGA were still sparse (Fig 1A), but their location extended up into the crypt middle, away from the SC niche (SC niche = crypt levels 1–20). Among adenomas, fewer than one-half of tumors showed any CGA staining. Nevertheless, in those adenomas that did show CGA staining, the population of stained cells within crypts extended even farther up the crypt (Fig 1A). Colon carcinomas showed few CGA+ cells.

Staining indices were also done to quantify the changes in crypt cell distribution of CGA + and ALDH+ cells in FAP tissues. The staining index for normal-appearing FAP crypts (Fig 1D) showed that the maximum proportion of cells that stained positively for CGA occurred at the crypt bottom, as for normal crypts, but CGA+ cells extended farther up the FAP crypt. For normal-appearing FAP crypts, the index curve for CGA no longer overlapped with the index curve for ALDH (Fig 1D). Rather, the ALDH curve was shifted upwards, indicating that there were more ALDH+ cells than CGA+ cells (a ratio of 1.6/1 compared to 1/1 for normal colon).

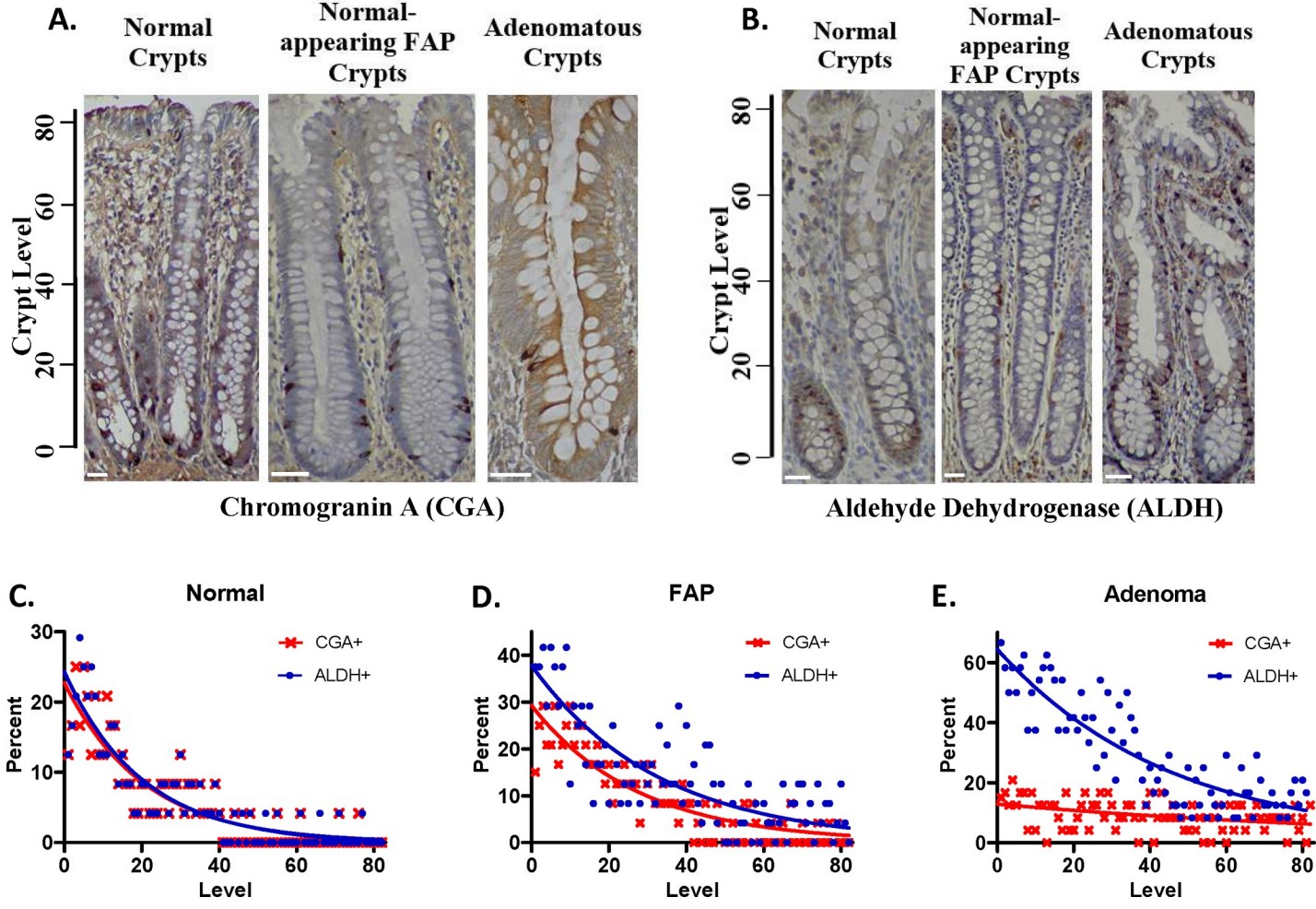

**Fig 1. The proportion of ALDH+ cells progressively increases while CGA+ cells markedly decreases in *APC*-mutant colon tissues from FAP patients.** Representative IHC stains of normal, normal-appearing FAP, and FAP adenomatous colonic crypts are shown for the expression of the: **(A)** NEC marker chromogranin-A (CGA). **(B)** SC marker ALDH1. Indices for CGA based on IHC mapping were plotted from staining of: normal crypts **(C)**, normal-appearing FAP crypts **(D)**, and FAP adenomatous crypts **(E)**. Quantitative IHC mapping was done on 24 full-length crypts for each index by scoring the number of positively stained cells as a function of crypt level. Indices for CGA are juxtaposed with ALDH1 indexes (from [23]). Results show that, in normal crypts, the distribution and proportion of cells staining for ALDH1 and CGA are very similar. In contrast, in *APC*-mutant tissues from FAP patients, the proportion of ALDH+ cells progressively increases while CGA+ cells markedly decreases. Crypt cell levels are indicated that range from crypt bottom (level 1) to crypt top (level ~82). Scale bars = 50 um.

The staining index for adenomatous crypts (Fig 1E) also showed a maximum at the crypt bottom, but in adenomas, the population of CGA+ cells extended even farther up the crypt than they did in FAP crypts. For adenomatous crypts, there was even greater separation of the two curves, with the ALDH curve shifted even higher relative to FAP crypts (Fig 1E). This was paralleled by a greater number of ALDH+ cells compared to CGA+ cells (a ratio of 3.25/1). Indices for carcinomas could not be constructed because carcinomas lack crypt structures.

### Immunofluorescence co-staining analysis shows that the frequency of co-expression of ALDH and CGA progressively decreases during colon tumorigenesis in FAP patients

**Normal crypts (homozygous wildtype APC).** To determine whether or not CGA and ALDH are expressed in the same cells, we used immunofluorescence to do co-staining (Fig 2).

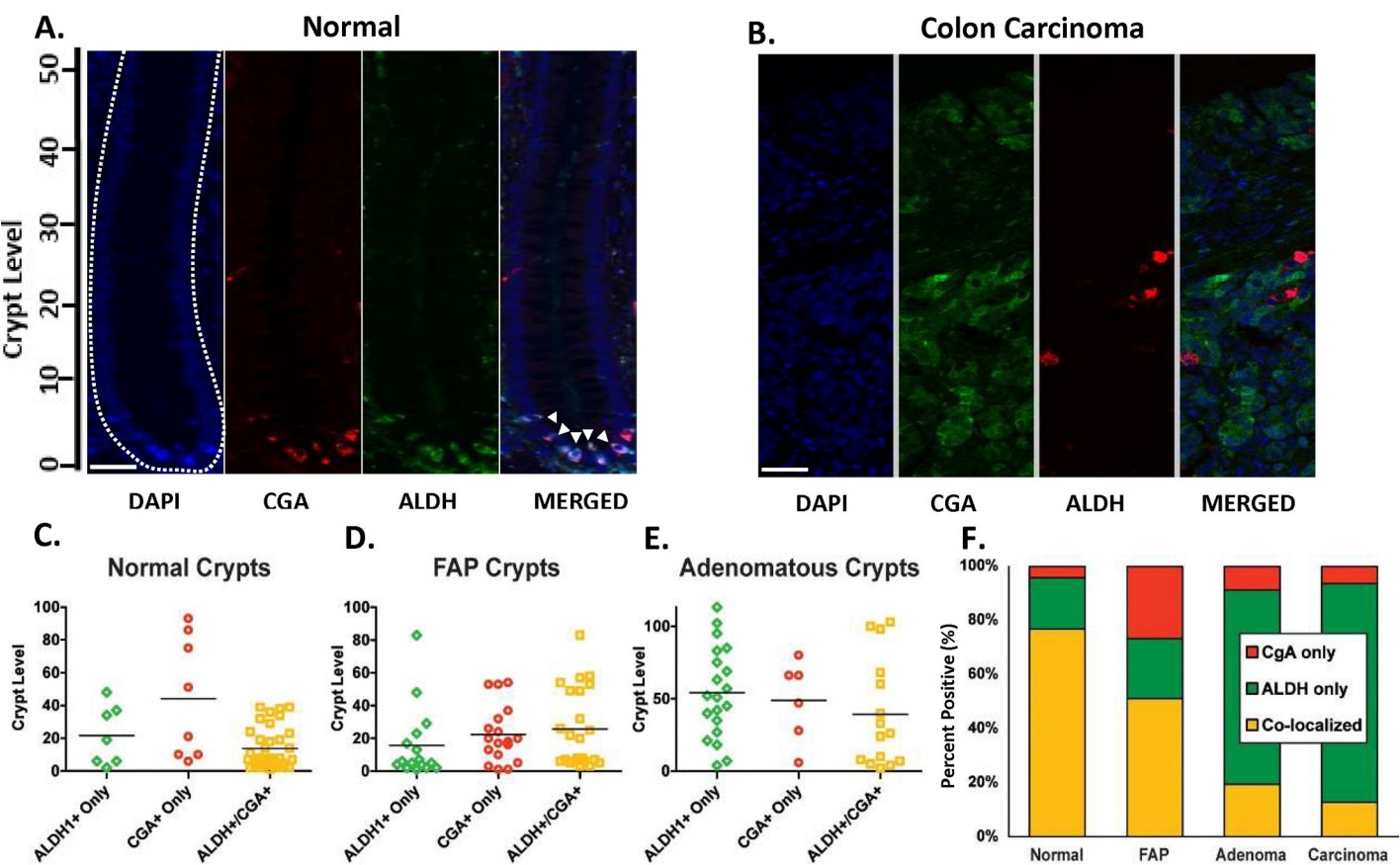

**Fig 2. The frequency of co-expression of ALDH and CGA progressively decreases during colon tumorigenesis in FAP patients.** Immunofluorescence co-staining studies were done to determine if CGA and ALDH are expressed in the same cells in normal and FAP colonic tissues. **(A)** Immunofluorescence microscopy images showing normal crypt cells staining for DAPI nuclear stain, CGA alone, for ALDH1 alone and merged for co-staining of both CGA and ALDH1. Crypt shape is outlined by the dashed line. **(B)** Immunofluorescence microscopy images showing staining of CRC tissues for DAPI nuclear stain, CGA alone, for ALDH1 alone, and merged for both CGA and ALDH1. Fig 2B shows that CGA and ALDH1 stains don't overlap in CRC. S1 Fig in S1 File shows representative images of co-staining for ALDH and CGA in colonic tissues with different *APC*-zygosity states. Staining of full-length crypts was also done to quantify the crypt location and proportion of cells staining positively for both CGA and ALDH1 and cells positive for CGA alone or ALDH1 alone. The crypt location of stained cells was analyzed in **(C)** normal, **(D)** normal-appearing FAP, and **(E)** adenomatous crypts. The proportions of ALDH+/CGA–, ALDH–/CGA+, and ALDH+/CGA+ cells is given in the stacked bar graph **(F)**. The number of positively stained cells counted in each case included, normal crypts (n = 95), normal appearing FAP crypts (n = 90), FAP adenomatous crypts (n = 103), FAP colon carcinomas (n = 126). Fig 2F shows that in normal crypts, many cells staining for the colonic SC marker ALDH1 co-stained for chromogranin-A (CGA) and other NEC markers. In contrast, in *APC*-mutant tissues from familial-adenomatous-polyposis (FAP) patients, the proportion of ALDH+/CGA+ SCs progressively increased while ALDH+/CGA–PNCs, and ALDH+/CGA–NECs progressively decreased. Crypt cell levels are indicated that range from crypt bottom (level 1) to the crypt top (level ~82). Scale bars = 50 um.

We found that most CGA+ cells also stained positively for ALDH1 (Fig 2A). Among those cells that stained positively for CGA or ALDH or both, about 77% co-stained for both markers. Among CGA+ cells, 95% co-stained for ALDH1. Among ALDH+ cells, ~80% stained positively for CGA.

In a separate analysis, we evaluated the crypt position of immuno-stained cells that were positive for both CGA and ALDH1, and cells positive for CGA alone or for ALDH1 alone. In this analysis, we mapped crypt level position of positively stained cells using only full-length crypts to determine the crypt cell location. In normal colonic tissue, most ALDH1+ cells were located in the bottom third of the crypt (Fig 2C–double-staining scatterplot). Cells that stained for both ALDH1 and CGA had a distribution similar to that of cells that stained for ALDH1 alone. We also analyzed 12 other markers for NECs (Table 1). Crypt cells staining positively for most of these other NEC markers were also located in the lower part of the colonic crypt.

**Table 1. Immunohistochemical analysis of normal colonic crypts for neuroendocrine markers.**

| Markers | Symbol | Crypt region showing the greatest number of positive cells | Proportion of crypt cells |
|---|---|---|---|
| Chromogranin A | CHGA | Bottom 1/3 | Few (~5%) |
| Syntaxin | STX | Bottom 1/3 | Few |
| Glucagon-like peptide 1 | GLP1 | Bottom 1/3 | Very few (<CGA) |
| Glucagon-like peptide 2 | GLP2 | Bottom 1/3 | Very few (<CGA) |
| Somatostatin | SST | Bottom 1/2 | Very few (<CGA) |
| Peptide YY | PYY | Bottom and middle 1/3 | Very few (<CGA) |
| Serotonin | 5-HT | Bottom 1/3 | Few |
| Neurotensin | NTS | no staining of crypts, only of stromal or neural cells | Scattered or nested |
| Synaptophysin | SYP | bottom 1/10 | Few |
| Glucagon | GCG | bottom 1/3 of the crypt (upper part) | Few |
| Neuron Specific Enolase | Γ-enolase | bottom (but not very bottom) | Few |
| Glucagon Receptor | CGCR | bottom 1/3 of the crypt (upper part) | Very few (<CGA) |
| Somatostatin receptor 1 | SSTR1 | Bottom 1/2 | Few |

In adenomas and carcinomas, the number of cells positive for these NE markers decreased. In adenomas and carcinomas very few cells stained for markers of NEC such as glucagon, GLP-1 and GLP-1R, somatostatin, or serotonin. All NE markers co-stained with CGA. Antibodies were from the following sources: Santa Cruz Biotechnology Inc, (Dallas, TX): Syntaxin 1 (1:100; #sc-73098), Glucagon-like peptide 1 (1:200; #sc-73508), Glucagon-like peptide 2 (1:200; #sc-7781), Somatostatin (1:300; #sc-55565), Peptide YY (1:250; #sc-80499), Neurotensin (1:300; #sc-377503), Synaptophysin (1:300; #sc-17750), Glucagon (1:250; #sc-514592); Abcam (Cambridge, MA): Chromogranin A (1:1000; #ab15160), Serotonin (1:250; #ab6336); Novus Biologicals (Centennial, CO): Neuron Specific Enolase (1:10,000; #NB-110-58870); R&D Systems (Minneapolis, MN): GLP-2R (1:100; #MAB4285); Advanced Targeting Systems (Carlsbad, CA): Somatostatin receptor 1 (1: 100; #AB-N35); BD Transduction Labs (San Jose, CA), ALDH1 (1:300; cat # 611195).

The crypt distribution of these other NEC types (noted in Table 1) was similar to that of CGA + cells (shown in Fig 1C).

**FAP patient tissues (APC mutant crypts).** We then determined, using immunofluorescence microscopy (evaluating only full-length crypts), the crypt location of cells positive for both CGA and ALDH1 and for cells positive only for ALDH1 or only for CGA in normal versus FAP colon tissues (Fig 2C–2E). For example, compared to normal tissue, where most ALDH1+/CGA−cells were located in the bottom third of the crypt, in adenomas the distribution of ALDH1+/CGA−cells extended farther up the crypt (Fig 2E). In all three tissues (normal, FAP, and adenomatous crypts) ALDH1+/CGA+ cells had a similar distribution as ALDH1+/CGA−cells. From normal to FAP to adenomatous crypts, the distribution of ALDH1+/CGA+ cells extended progressively farther up the crypt.

To further evaluate changes in NEC populations during the progression to colon cancer, we examined the expression of the other 12 NEC markers (Table 1). In adenomas and carcinomas, the number of cells staining positively for these NE markers decreased, paralleling the decrease in CGA staining.

We also evaluated the proportion of CGA/ALDH double-stained cells in FAP neoplastic tissues (Fig 2F). While double-staining of FAP tissues, including CRCs (Fig 2B), showed that some CGA+ cells co-stained for the SC marker ALDH1, and vice versa, there was a substantial decrease in the frequency of co-staining (normal > FAP > adenoma > carcinoma). Compared to normal tissue, where 77% of cells surveyed co-stained for CGA and ALDH1, in normal-appearing FAP crypts, the proportion was only 51%; in adenomatous crypts it was 19%; in carcinomas it was 13% (Fig 2F). The number of cells that stained for ALDH1, but not CGA, showed a trend in the opposite direction with proportions being, respectively, 19%, 22%, 72%, and 81%.

## Immunofluorescence co-staining analysis shows that distinct subpopulations of NECs having different phenotypes exist in the human colonic SC niche

We also determined if CGA and other markers for NECs are expressed in the same cells using co-staining analysis of normal crypts (Fig 3). Many but not all cells staining positively for CGA co-stained for each of these other markers (and vice versa, Fig 3A). For example, cells staining for CGA also stained for PYY, serotonin, somatostatin, GLP-1 or GLP-2 (Fig 3A). Most syntaxin-positive cells stained for CGA and vice versa and the number of cells that were positive for CGA and syntaxin was greater than the number of cells staining for the other NEC markers (Table 1). Double-staining for glucagon (a stimulator of crypt proliferation) and somatostatin (an inhibitor of crypt proliferation) showed that cells that stain for glucagon only do not stain for somatostatin and vice versa (Fig 3A, lower right panel).

We also did immunostaining of cells for neuropeptides and their respective receptors including glucagon and glucagon receptors and somatostatin and the somatostatin receptor SSTR-2. The results (Fig 3B) show that expression of these neuropeptides and their respective receptors are co-localized.

## Mathematical modeling of crypt cell kinetics in normal and neoplastic colon indicate *APC* mutations lead to decreased maturation of ALDH+ SCs into progenitor NECs (not progenitor NECs into mature NECs)

To identify kinetic mechanisms that underlie how the proportion of SCs, PNCs, and mature NECs change from normal to FAP to adenoma to CRC tissues, we used mathematical modeling. Modeling output indicated a significant increase occurred in the renewal probability of SCs as tumorigenesis progressed (Fig 4A). Specifically, in modeling cell dynamics in CRC, the probability that a cancer SC will self-renew was calculated to be 91.8% compared to 60% for a normal SC. Our modeling results also showed that the division rate of SCs decreased in the colonic epithelium during the progression from normal to cancer. In CRC, cancer SCs were calculated to divide at half the rate of normal SCs (Fig 4B). We also observed that the computed death rate for NECs in a normal crypt was more than 10 fold greater than that for neoplastic tissues (Fig 4C). This was attributed to the high PNC number relative to the low NEC number for normal crypts.

We also investigated the number of SCs coming in via self-renewal minus the number of SCs going out via cell division from the SC compartment (defined as the reproduction number). Based on our modeling, SC reproduction increased 1.4 fold in FAP, 2.5 fold in adenomas, and 2.2 fold in CRCs compared to normal colon (Fig 4D). Overall, these computational analyses indicate that: 1) FAP tissues display a progressive decrease in the maturation of immature SCs (ALDH+/CGA− cells) into PNCs (ALDH+/CGA+ cells), but not decreased maturation of PNCs (ALDH+/CGA+ cells) into mature NECs (ALDH−/CGA+ cells); 2) feedback signaling from mature NECs (ALDH+/CGA− cells) is diminished (Fig 4E).

## Biological experiments on effects of NEC signaling on ALDH+ cells in CRC cell lines show feedback regulation of progenitor NECs occurs via mature GLP-2R+ and SSTR1+ NECs

To test model predictions, we determined if NECs regulate maturation of SCs along the NEC lineage by analyzing how SSTR1+ and GLP-2R+ cells affect the proliferation and sphere-forming ability of isolated ALDEFLUOR+ cells from CRC cell lines. Because our immunostaining results showed that colonic cells do not co-stain for SSTR1 and GLP-2R (Fig 3G–3J),

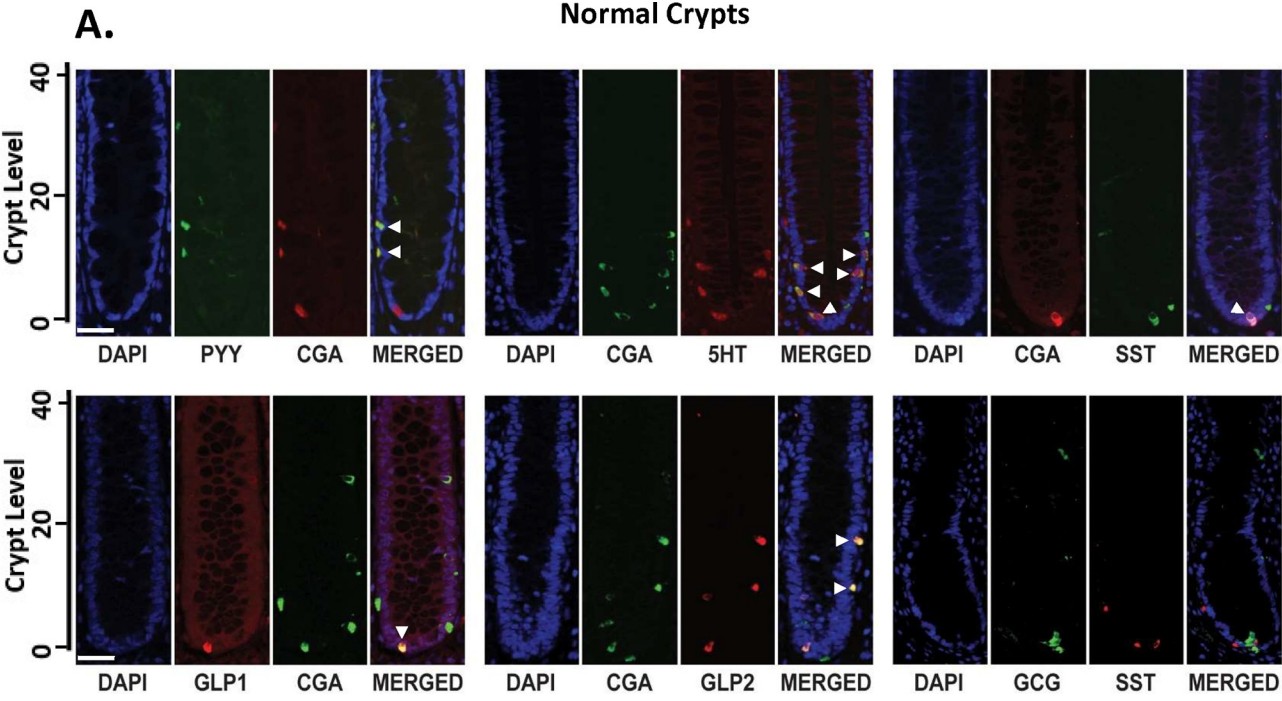

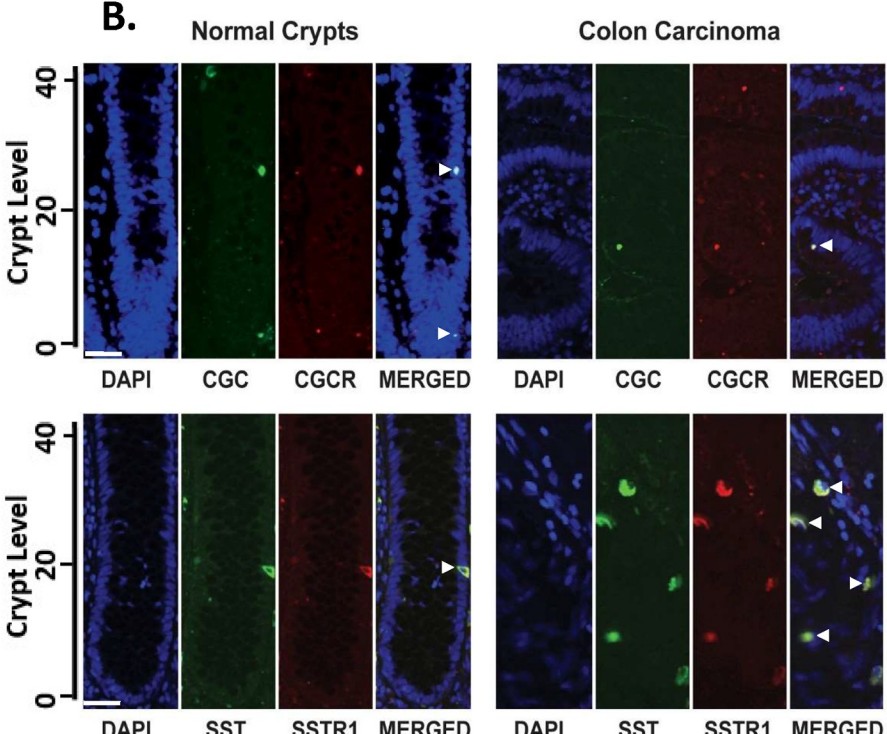

**Fig 3. Co-staining shows SSTR1+ and GLP-2R+ cells are distinct NEC sub-types that selectively express SST and GLP-2, respectively.** Co-staining was done for expression of NEC markers in normal crypts. The following co-staining patterns in normal crypts are shown in Fig 3A panels: (upper right) PYY/CGA; (upper middle) 5-HT/CGA; (upper right) SST/CGA; (lower left) GLP-1/CGA; (lower middle) GLP-2/CGA; (lower right) GLP-2/SST. Results in Fig 3A show that: (i) many, but not all cells, staining positively for CGA co-stained for each of these other NEC markers (and vice versa); (ii) cells staining for GLP-2 do not stain for SST and vice versa. The following co-staining patterns in normal crypts and colon carcinomas are also shown in

Fig 3B: GLP-2/ GLP-2R and SST/SSTR1 (see Table 1 for abbreviations). Results in Fig 3B show that in both normal crypts and colon carcinomas, SSTR1 + and GLP-2R+ cells are distinct NEC sub-types that selectively express their own neuropeptides SST and GLP-2. Crypt cell levels are indicated that range from crypt bottom (level 1) to the crypt top (level ~82). Scale bars = 50 um.

indicating that these cells represent different NEC types, we first screened different CRC cell lines (n = 5) and characterized them in terms of proportion of ALDEFLUOR+, SSTR1+ and GLP-2R+ cells. Results showed that all CRC lines analyzed had sub-populations of ALDE-FLUOR+, SSTR1+ and GLP-2R+ cells (Fig 5A).

To elucidate the effect of exogenous SST and GLP-2 on ALDEFLUOR+ sub-population size, HT29 cells and SW480 cells were treated with the IC50 dose of GLP-2 (500 nM) or of SST (500nM) for 24 hours. Results showed that treatment with exogenous GLP-2, but not SST, significantly decreased the proportion of ALDH+ cells in both cell lines as determined by ALDE-FLUOR assay (Fig 5B and 5C).

We then investigated whether paracrine signaling from NECs might regulate ALDH+ cells isolated from the HT29 CRC cell line. Accordingly, conditioned medium (CM) was collected from GLP-2R+ or from SSTR1+ cell cultures and ALDEFLUOR+ cells were then grown in this CM for one week. We found that GLP-2R+ CM, but not SSTR1+ CM, significantly decreased the ALDEFLUOR+ population size as compared to the mock-sorted control medium (Fig 5D).

To evaluate for juxtacrine signaling, co-culture experiments were then done to study the effect of direct cell-cell contact between NECs and ALDH+ cells isolated from the HT29 cell line. Because we observed that co-cultures of ALDEFLUOR+ cells with GLP-2R+ cells, but not with SSTR1+ cells, increased the total number of cells in culture after 1 week, we determined the effect of the co-culturing on the proportion of ALDH+ cells. We found that co-cultures of ALDEFLUOR+ cells with SSTR1+ cells, but not with GLP-2R+ cells, showed an increased percentage of ALDEFLUOR+ cells (Fig 5E).

To study the effect of SST signaling on the maturation of ALDH+ cells into different cell types, we evaluated the cell sub-populations in spheres that were generated from ALDE-FLUOR+ cells isolated from the HT29 line. Accordingly, ALDEFLUOR+ cells isolated from the HT29 line were grown in presence of SST or SST inhibitor (cycloSST) under ultra-low attachment sphere-forming conditions for 10 days. We found the proportion of ALDEFLUOR + cells was high in spheres formed in presence of SST and low in presence of cycloSST (Fig 5F). Both SST and cycloSST decreased the proportion of SSTR1+ cells (Fig 5G). We also found that expression of the MCM2 proliferative cell marker in spheres was decreased in presence of SST and increased in presence of cycloSST (Fig 5H).

## Discussion

A key finding of our study using quantitative immunohistochemical mapping of human colonic tissues is that for normal crypts, staining indices for CGA and ALDH1 are virtually superimposable (Fig 1). This indicates that CGA+ and ALDH1+ cells share the same niche (at the crypt bottom) and account for similar proportions of cells in that niche. An even more striking finding that the superimposition suggested was that the two markers were labelling many of the same cells. Indeed, co-staining experiments showed that ~80% of ALDH1+ cells co-stain for CGA and vice versa. Thus, many ALDH1+ cells in the normal colon express a NEC phenotype and even ALDH1+/CGA+ co-staining cells are likely to be cells with high stemness (see [99] for a discussion of stemness). A plausible explanation for why ALDH1 + cells have a NEC phenotype is that the cells that express only ALDH1 (i.e., ALDH1+/CGA– cells) are the true immature SCs. While cells that express both ALDH1 and NE markers

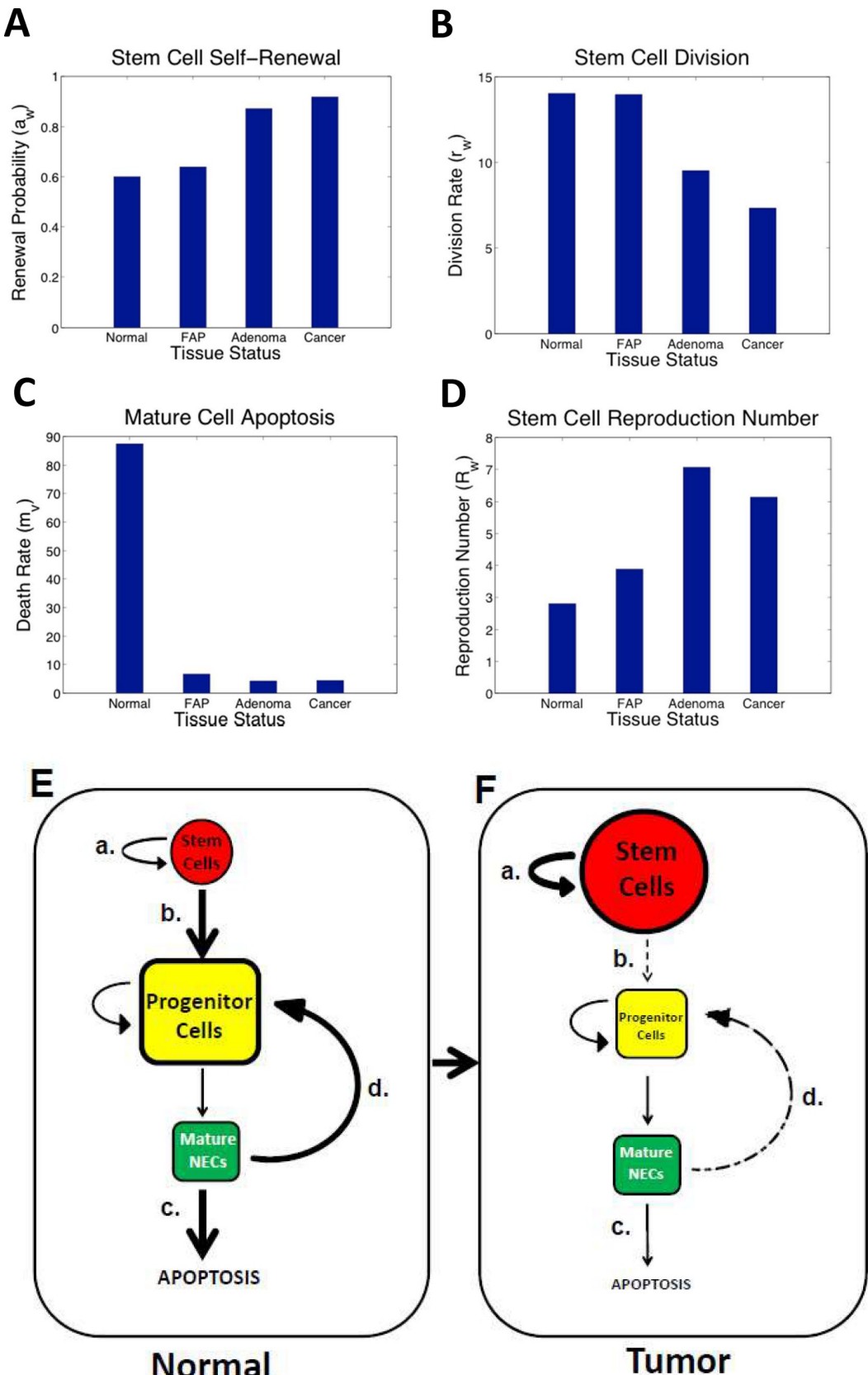

**Fig 4. Mathematical modelling indicates that *APC* mutations cause decreased maturation of ALDH+ SCs into progenitor NECs.** Mathematical modeling results provide an explanation for neuroendocrine cell kinetics in normal and neoplastic colon. Model output indicated the following: **(A)** A significant increase (61.8% to 91.8%) in SC self-renewal probability as tumorigenesis progressed from normal to FAP to adenoma to cancer. **(B)** The division rate of SCs declined (from 14 to 7.3). **(C)** The rate of death (apoptosis) of mature cells in a normal crypt was an order of magnitude greater than in neoplastic tissues. **(D)** The stem cell reproduction number ($R_w$) is increased in *APC*-mutant tissues. The modeling results also indicated that the division rate of SCs decreased in colonic crypts during the progression from normal to cancer as described in the S1 File. The diagram in **(E)** depicts the relative size (normal vs. tumor) of SC (red), PNC (yellow), and NEC (green) compartments, which illustrates how altered reactions involved in maturation of SCs along the NEC lineage give rise to changes in the size of the SC, PNC, and NEC populations in FAP crypts. Modelling results showing a lower rate of SC division **(B)** in neoplastic tissues indicates that there is diminished feedback regulation of PNCs by mature NECs, and a delay in maturation along the NEC lineage **(E** vs **F)**. Model output showing increased probability of SC self-renewal **(A)** and increased SC reproduction number **(D)** is consistent with progressive SC overpopulation occurring during colon tumorigenesis in FAP patients.

(ALDH1+/CGA+ cells) are the PNCs maturing along the NEC lineage, which retain some degree of stemness. In this view, ALDH–/CGA+ cells would be fully mature NECs, which is the final stage of SC maturation along the NEC lineage.

A second key finding is that not only do PNCs co-express SC and NEC markers, but also, they are found in the same location (levels 1–20) as are immature SCs (Fig 2C). This suggests that both cell types reside in the SC niche. Thus, these data support our first hypothesis: that within the SC niche of normal human colonic crypts, SCs mature along the NEC lineage.

Third, our results show that in neoplastic colonic epithelium, the frequency of co-staining monotonically decreases with tumor progression: 32% less (than normal) in FAP crypts; 74% less in adenomatous FAP crypts; 83% less in carcinomas (Fig 2F). Similarly, in CRC cell lines, the proportion of ALDH+ cells that co-stain for NEC is negligible and the proportion of ALDH+ cells is greater than that of NECs (GLP-2R+ and SSTR1+ cells; Fig 5A). These data provide evidence for our second hypothesis: that with sequential mutational inactivation of *APC* genes in neoplastic crypts, SC maturation along the NEC lineage becomes progressively delayed.

Fourth, our results show that ALDH1+/CGA+ and ALDH1+/CGA–cells are anatomically co-localized in mutant crypts as they were in normal crypts (Fig 2D and 2E). This co-localization provides further evidence that both normal and *APC* mutant colonic SCs mature along the NEC lineage. Moreover, when the distribution of ALDH+/CGA–cells expanded upward toward the crypt top in mutant crypts, the distribution of ALDH+/CGA+ cells also expanded upward toward the crypt top. This upward expansion, in neoplastic crypts, of the populations of both ALDH1+/CGA+ and ALDH1+/CGA–cell types is precisely what one would predict if there is delayed maturation of SCs along the NEC lineage due to *APC* mutations. This provides further support for our second hypothesis.

A fifth finding is that the number of ALDH1+/CGA–cells increases during colon tumorigenesis, which is based on two observations. First, staining indices showed that, during colon tumorigenesis, there was a progressive increase in the size of the ALDH+ cell population relative to the size of the CGA+ population (Fig 1C–1E). Second, double-staining (Fig 2F, bar graphs) showed a monotonic decrease, with tumor progression, in the proportion of ALDH1+ cells that co-stain for CGA, and a roughly equal and opposite change, a monotonic increase, in the proportion of ALDH1+ cells lacking CGA staining. Taken together, the indices and double-staining data indicate that there is an increase in the absolute number of ALDH+/CGA–cells during colon tumorigenesis. This increase is tantamount to SC overpopulation and is consistent with evidence from our previous mathematical modelling [20] and biological studies [22, 23] indicating that SC overpopulation occurs during colon tumorigenesis. Other studies using various SC markers also suggest that SC overpopulation occurs during intestinal

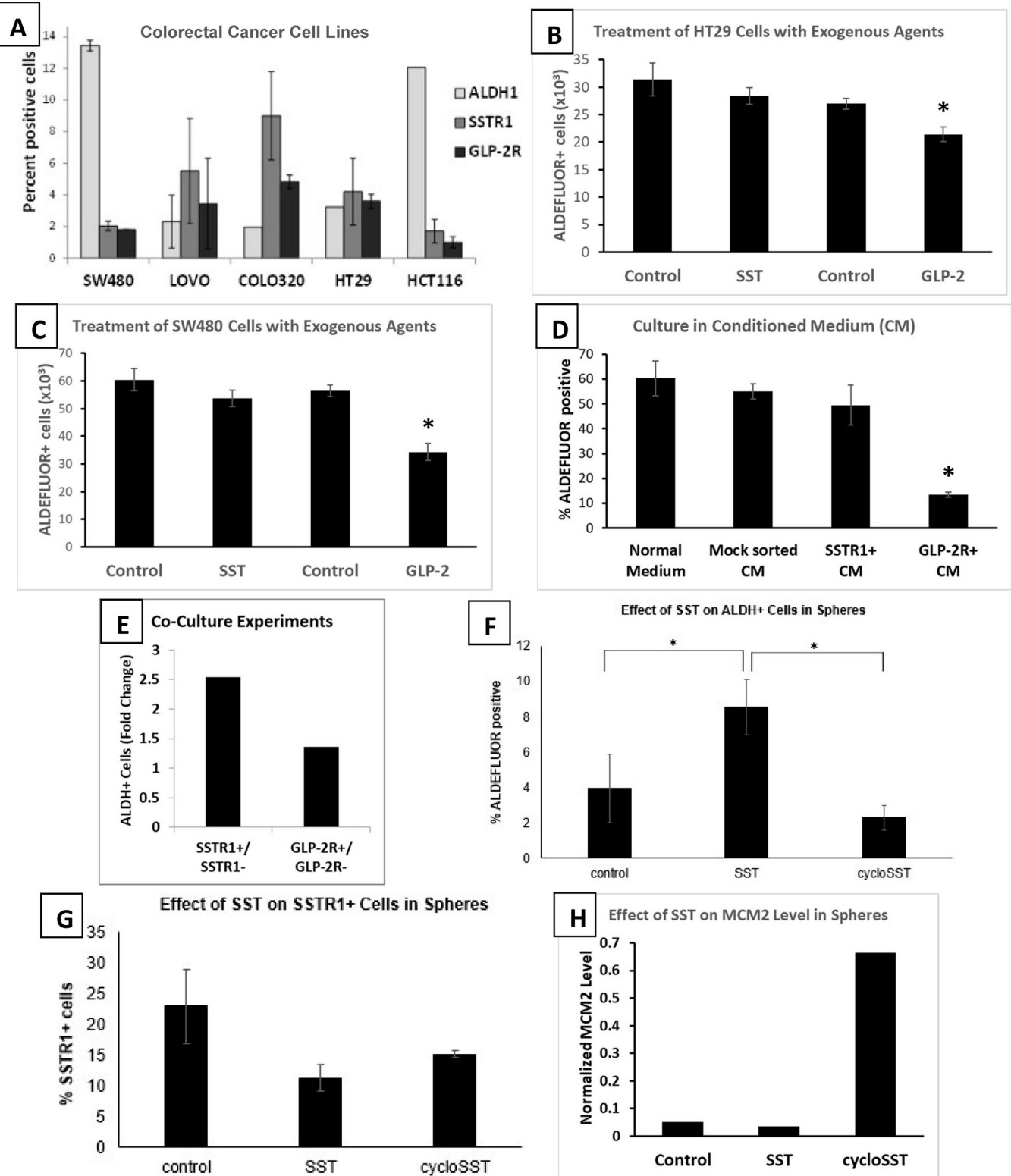

**Fig 5. Biological experiments using human CRC cell lines show feedback regulation of progenitor NECs occurs via mature GLP-2R+ and SSTR1+ NECs.**
Fig 5 shows analysis of the effects of SST and GLP signaling on ALDH+ stem cells in CRC cell lines. **(A)** *ALDEFLUOR+, SSTR1+, and GLP-2R+ cells in CRC cell lines.* To investigate how NECs might be involved in regulation of ALDH+ cells we analyzed the proportion of ALDEFLUOR+, SSTR1+, and GLP-2R+ cells in CRC cell lines. Results showed that all CRC lines analyzed have sub-populations of ALDEFLUOR+, SSTR1+, and GLP-2R+ cells. **(B & C)** *Effect of exogenous*

*SST and GLP-2 on ALDH+ population size in CRC cell lines*. Treatment of HT29 and SW480 cell with GLP-2, but not SST, significantly decreased the percentage of ALDH+ cells. Error bars are ±SE; * = p<0.05. **(D)** *Culture of ALDH+ cells in conditioned medium (CM)*. To investigate paracrine signaling influence of NECs on ALDH+ cells, we studied the effect of CM from GLP-2R+ or from SSTR1+ cells on ALDEFLOUR+ cells isolated from the HT29 cell line. There was a significant decrease in the number of ALDEFLOUR+ cells grown in GLP-2R+ CM, but not SSTR1+ CM. Error bars are ±SE; * = p<0.05. **(E)** *Co-culture experiments to investigate the effect of cell-to-cell contact from SSTR1+ and GLP-2R+ NECs on ALDH+ cells*. Co-culture experiments were done using ALDEFLUOR+, SSTR1+, SSTR1−, GLP-2R+, and GLP-2R− cell subpopulations isolated from the HT29 cell line. Changes in ALDH+ cells are expressed as the ratio of number ALDH+ when co-cultured with SSTR1+ cells versus SSTR1− cells as well as when co-cultured with GLP-2R+ cells versus GLP-2R− cells. Co-cultures of ALDEFLUOR+ cells with SSTR1+ cells, but not with GLP-2R+ cells, showed an increased proportion of ALDH+ cells relative to co-culture with SSTR1− and GLP-2R− cells, respectively. **(F, G, H)** *Effect of SST on SSTR1+ cells, ALDH+ cells, and on expression of the MCM2 proliferative cell marker in colonospheres*. The effect of SST on cell subpopulations in colonospheres, was tested in colonosphere assay for 10 days using colonospheres grown from ALDEFLUOR+ cells isolated from the HT29 cell line. **(F)** SST increased and cycloSST decreased the proportion of ALDEFLUOR+ cells in colonospheres. Error bars are ±SE; * = p<0.05. **(G)** Both SST and cycloSST decreased the proportion of SSTR1+ cells in colonospheres. Error bars are ±SE. **(H)** SST decreased and cycloSST increased expression of the MCM2 proliferative cell marker in colonospheres. The effect of SST on expression of the MCM2 proliferative cell marker in ALDEFLOUR+ cell-generated colonospheres that formed in presence of SST or cycloSST was determined by western blotting analysis and results were quantified by densitometry. S2A-S2C Fig in S1 File shows representative flow cytometry data from these experiments.

tumorigenesis in humans as well as in mice [25–33, 42]. Taken together, these findings indicate that delayed maturation (due to *APC* mutations) along the NEC lineage contributes to SC overpopulation during CRC development. Several other important points bear further discussion.

## Do mouse intestinal SCs also express a NEC phenotype?

Because we found that many cells in normal human colonic crypts staining for the colonic SC marker ALDH1 co-stained for chromogranin-A (CGA) and other NEC markers, we asked if mouse intestinal SCs also express a NEC phenotype. As noted in the introduction, although early studies showed that ALDH marks murine intestinal SCs, ALDH was not extensively pursued as a marker for intestinal SCs in mice. As it turns out, however, murine intestinal SCs identified by several other SC markers do show that they have a NEC phenotype. A chronology of these investigations on mouse intestine is as follows. In 2011, Sei et al. [100] found that CCK + NECs in the crypt SC niche (below +4 position) were positive for the several SC markers, including Lgr5, CD133, and doublecortin and CaM kinase-like-1, as well as for the neuroendocrine transcription factor neurogenin 3. In 2013, Buczacki et al. [101] observed that mouse intestinal label-retaining cells are secretory precursors expressing Lgr5 that are committed to mature into differentiated secretory cells of Paneth and enteroendocrine lineages. Moreover, they found that after intestinal injury these cells were capable of extensive proliferation and gave rise the main epithelial cell types. In 2015, Hayakawa et al. [102] identified secretory precursor (BHLHA15+) cells in mouse intestine and colon. These BHLHA15+ cells have SC-like properties and can supply the enterocyte lineage after injury. In 2016, Sasaki et al. [92] showed that Reg4+ deep crypt secretory cells in mouse intestine have a NEC phenotype. In 2017, Basak et al. [98] reported that induced quiescence of Lgr5+ SCs in intestinal organoids enabled differentiation of hormone-producing NECs. Also, in 2017, Yan et al. [103] observed that in murine crypts, Bmi1+ SCs were distinct from Lgr5+ SCs. And, Bmi1+ cells, but not Lgr5+ cells, were enriched for several NEC markers, including Prox1. Moreover, a lineage-tracing of Prox1+ cells indicated that the NEC lineage (including mature NEC cells) comprises a reservoir of long-lived clones during homeostasis and regeneration of intestinal SCs after radiation-induced injury. In 2019, van Es et al [104] reported that after ablation of Paneth cells in murine crypts, NECs can replace Paneth cells to support Lgr5+ SCs. Overall, these studies show that murine intestinal SCs have a NEC phenotype (particularly quiescent, Bmi1+ cells) which provides credence to our conviction that human colonic ALDH+ cells that display a NEC phenotype are in fact PNCs with high degree of stemness. Indeed, as discussed below, our biological experiments indicate that these PNCs display the ability for self-renewal in sphere formation assays.

## Why do SC, PNC, and NEC populations change in FAP tissues?

We conducted mathematical modelling to identify kinetic mechanisms that might explain how the changes in SC, PNC, and NEC populations occur in FAP tissues. Our computational analyses showed a significant increase in the renewal probability for ALDH+/CGA−cells, but not for ALDH+/CGA+ cells in FAP tissues during CRC development. This finding indicates that *APC* mutations cause decreased maturation of immature SCs into PNCs, but not the maturation of PNCs into NECs. Modelling results also indicated that mature NECs (ALDH−/CGA+) regulate the rate of cell maturation along the NEC lineage through a feedback mechanism. This raised the possibility that feedback regulation, through production of neurotransmitters by mature NECs, might be involved in controlling the maturation of SCs along the NEC lineage and that diminished feedback regulation might contribute to a delay in NEC maturation in FAP tissues. We then tested this possibility using biological experiments.

## How might neuroendocrine signalling play a role in NEC maturation?

We conducted biological experiments using human CRC cell lines to determine if NE factors might have a role in regulation of NEC maturation. That ALDH1+/CGA+ cells express NE factors suggests that regulation could occur through an autocrine or intracrine mechanism in which the NEC would be regulating its own maturation as it matures [105, 106]. Indeed, our study shows that SSTR1+ and GLP-2R+ cells produce their own corresponding ligands in normal crypts as well as in colon carcinomas (SST and GLP-2, respectively; Fig 3G–3J). Alternatively, that smaller subpopulations of ALDH−/CGA+ cells exist in normal crypts, but don't co-stain, also suggests the possibility that paracrine or juxtacrine mechanisms may be operative. Indeed, this prospect is supported by our finding that GLP-2R and SSTR1 ligands (Fig 5B–5E; [38]) can modulate the growth of ALDEFLUOR+ cells. Our idea that NE mechanisms regulate maturation of NEC populations is consonant with an earlier proposal that crypt NECs have a role in maintaining the SC sub-populations in the SC niche [85]. Of course, NECs may be regulated by outside factors such as the enteric nervous system, autonomic nervous system, and CNS (via the brain gut axis).

While the effect of NE factors on proliferation in the gut is well defined, less is known on how they are involved in regulation of maturation. For example, glucagon, a growth stimulator [93–95], and somatostatin, a growth suppressor [38, 107], are known opposing regulators of GI proliferation. Moreover, glucagon is known to be produced from type A($\alpha$) NECs and somatostatin from D($\delta$) NECs [108]. That these two peptides are produced in distinct NEC types and have opposing effects on proliferation suggests that they might have different effects on SC or PNC maturation. The current study evaluated this possibility. For example, in our study of colonic tissues and of CRC cell lines we found that SSTR1+ and GLP-2R+ cells represent distinct sub-populations. And, our *in vitro* experiments show that these two NE factors have opposite effects on ALDEFLUOR+ cells. For example, exogenous GLP-2, but not SST, decreased the ALDEFLUOR+ population size (Fig 5B & 5C; [38]). Moreover, conditioned medium from GLP-2R+ cells (that express GLP-2), but not SSTR1+ cells (that express SST), significantly decreased the ALDEFLUOR+ population size (Fig 5D). Taken together, our results reveal that not only are GLP-2 and SST produced in distinct NEC types, but also, these ligands have opposite effects on ALDEFLUOR+ cells. Thus, our results reveal that SST signaling reduces, and conversely, GLP-2 enhances the rate of maturation along the NEC lineage.

We performed additional experiments to evaluate the effect of NE factors on generation of NECs and proliferative cells in colonospheres formed from ALDH+ cells in ultra-low attachment cultures. We observed that spheres formed from ALDEFLUOR+ cells in the presence of SST showed a high proportion of ALDEFLUOR+ cells and low level of expression of the

MCM2 proliferative cell marker (Fig 5F and 5H), which indicates that SST reduces the rate of maturation of ALDEFLUOR+ cells. The SST inhibitor, cycloSST, had the opposite effect. That the size of the ALDEFLUOR+ population can be modulated by SST signaling might suggest that feedback mechanisms exist between mature NECs (ALDH–/CGA+ cells) and immature SCs (ALDH1+/CGA–cells). However, because ALDEFLUOR+ is based on ALDH enzymatic activity, it does not distinguish between ALDH1+/CGA–cells and ALDH1+/CGA+ cells. Moreover, based our findings showing that ALDH1+/CGA–cells do not have receptors for NE ligands, the feedback effects of mature SSTR1+ and GLP-2R+ NECs, via SST and GLP-2, must be on PNCs, not immature SCs (Fig 4E). That these neuropeptides modulate sphere formation shows PNCs (ALDH1+/CGA+ cells) have a high degree of stemness and a capacity for self-renewal.

## Could dysregulation of the feedback mechanism lead to delayed NEC maturation in CRC?

This question pertains to whether dysregulation of feedback between mature NECs and PNCs might be the mechanism that explains how the subpopulation of immature SCs (ALDH+/CGA–cells) selectively expands in size relative to the NECs (ALDH+/CGA+ and ALDH–/CGA+ cells) in FAP tissues. Delayed maturation is a mechanism that is well known for some hematopoietic cancers [109] where it is referred to as maturation arrest. Also, we reported (for both modelling studies and biologic studies) evidence for delayed maturation in colon tumorigenesis [20, 38, 39, 99, 110]. Thus, decreased rate of maturation of SCs along an NEC lineage would explain the relatively smaller size of the NEC population in colon tumors with mutant *APC*. Perhaps it is an alteration of feedback mechanisms that control the rate of NEC maturation that creates a maturation arrest or "bottleneck" effect, which is the mechanism that leads to SC overpopulation and contributes to the development of colon tumors. However, if this was the case, there would be a buildup or increase in PNCs at the maturation point of the arrest. In contrast, there is actually a decrease in the number of both ALDH+/CGA+ and ALDH–/CGA+ cells in FAP tissues, indicating that both PNCs and NECs become depleted in CRC development. Thus, dysregulation of a feedback loop between NECs and PNCs wouldn't explain the delayed maturation. This conclusion is also what our modeling predicted (Fig 4F) as discussed above. So the mechanism for delayed maturation must be connected to how *APC* mutations cause the immature ALDH+ SCs to remain immature.

In exploring this mechanism, we must also consider what it means for a SC to be ALDH positive. The presence of ALDH in NECs (ALDH1+/CGA+) cells, particularly as seen in non-malignant cells, suggests that ALDH plays a role in the maturation of SCs along the NEC lineage. Indeed, ALDH is known to have a role in early maturation of SCs by catalysing the production of retinoic acid from retinol [54–56]. And, decreased retinoic acid receptor signalling has been reported for colon cancers [111, 112]. This information, coupled with our findings in the current study, suggests that ALDH is required for the maturation of colonic SCs, in particular their ability to mature into NECs.

## How do *APC* mutations lead to decreased NEC maturation?

This question is first addressed by discussing how NEC maturation is distinct from maturation of other crypt cell types in the colon, which will help us recognize which signaling pathways are specific to regulating NEC maturation. Much is known about the various cell lineages in the intestinal crypt and the pathways that mediate maturation along these lineages [76, 77]. The prevailing view is that mutations in *APC* lead to decreased maturation across all lineages in the intestine. But, the mechanism by which *APC* mutation alters the specific pathways that

mediate maturation of each lineage is less well understood, particularly for NECs in human colon.

The maturation of NECs is unique because NECs have morphological and functional properties of both endocrine cells and neurons. Morphologically, unlike most crypt cells that are frustum-shaped, NECs are pyramidal-shaped, and contain neural synapse-like storage vesicles [81, 82]. Functional features shared with neurons include uptake, synthesis, storage and release of various neurotransmitters and neuropeptides [reviewed in 85]. Thus, the process of differentiation of NECs is distinct from the overwhelming majority of other cells within the crypt that are non-endocrine, particularly absorptive and goblet cells. Unlike these non-endocrine cells that differentiate as they migrate upward and proliferate along the crypt axis, NECs appear to differentiate directly from SCs located within the SC niche. Although they mostly reside in the niche, NECs have long basal neuron axon-like processes that extend along the basement membrane in the crypt [reviewed in 85]. Moreover, unlike absorptive and goblet cells that have a short turnover time, kinetic studies show that NECs have an extended lifespan [113–116]. Their prolonged lifespan is not only due to their slow proliferation rate, but also by the fact that NECs uniquely express collagen IV [117], which allows them to avoid adhering to and co-migrating with other cell lineages. NECs are also distinguished by their radio-resistance and ability to repopulate other cell lineages after radiation-induced injury, indicating they are resilient and have regenerating ability. Given these unique properties of NECs, it suggests that differentiation mechanisms that are altered by *APC* mutations are different from the other differentiated cell types in the colon.

Accordingly, we focus here on mechanisms that specifically may affect differentiation of NECs from ALDH+ SCs. Because ALDH is an enzyme that is key to retinoid acid (RA) signaling and retinoids are well known to promote differentiation of SCs [52], it follows that having ALDH in a SC provides the capacity for it to differentiate in response to retinoids. Indeed, we show that the key components of the retinoid signaling pathway are contained in ALDH+ SCs and that ATRA induces their differentiation along the NEC lineage [39]. This is consistent with the ability of retinoids to decrease proliferation of ALDH+ SCs and, conversely, that inhibitors of ALDH increase proliferation of ALDH+ SCs [51–54]. Moreover, WNT signaling upholds ALDH expression that may enable ALDH+ SCs to differentiate because ALDH1 is a TCF4 target gene [61, 65, 118–128]. Although, as discussed below, a decrease in WNT signaling appears necessary for the retinoid signaling to happen. Perhaps, a transient decrease in WNT signaling provides a window for retinoid signaling to occur due to oscillations in level of WNT signaling that occur, which we [129, 130] and others [131–133] have observed. Thus, it appears that *APC* mutations may alter the ability of ALDH+ SCs to differentiate in response to retinoids, which would lead to expansion of the ALDH+ SC population size in CRC.

Since *APC* mutations are known to increase WNT signaling in FAP, this raises the question: does increased WNT signaling lead to decreased retinoid signaling? To address this question we draw from research showing that retinoids induce differentiation of neural cells. Indeed, it is well known that retinoids induce neural differentiation of embryonic SCs, and that retinoid signaling is essential in the development of the central nervous system in most animal species [134–137]. Moreover, endogenous retinoic acid signaling is critical to guide the self-organization and pattern formation of neural tissue from mouse embryonic SCs [138, 139].

Key to our question—How does mutant *APC* decreases NEC differentiation?—It is important to point out that appropriately regulated WNT signaling is necessary for RA to induce neuronal differentiation [140]. For example, an inhibitor of the enzyme GSK3β activity that increases WNT signaling blocks the ability of ATRA to induce neural lineage differentiation [53]. Not only does WNT suppress retinoid signaling, but conversely, increased RA signaling suppresses WNT's ability to block retinoid induction of the neural differentiation of SCs. For

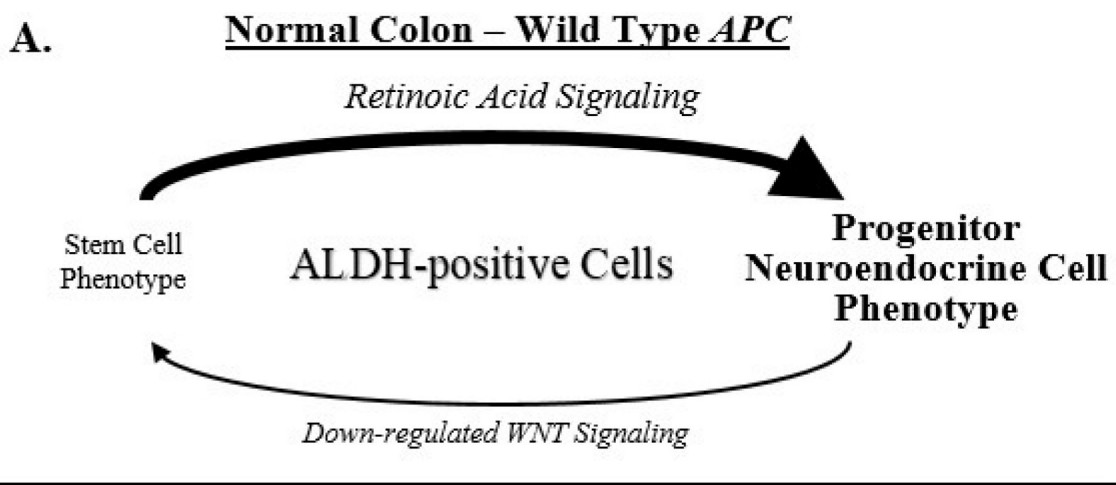

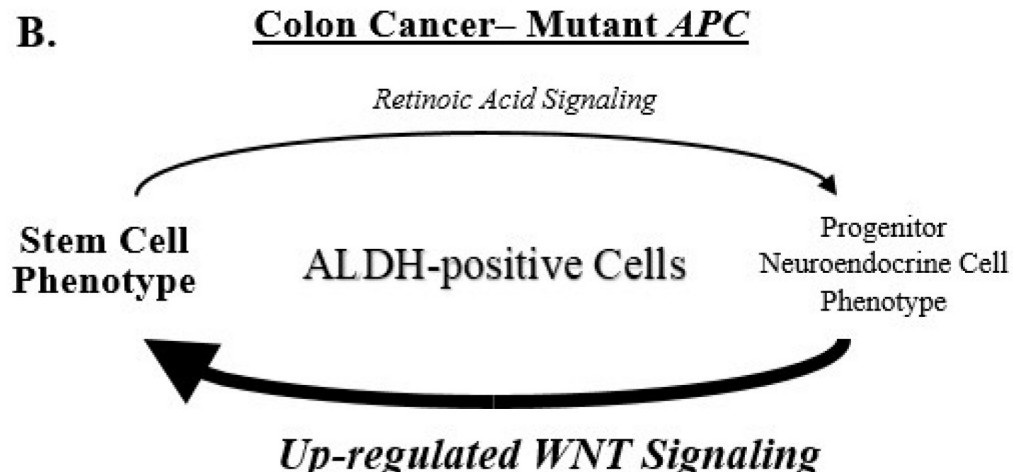

**Fig 6. Our proposed schematic for the attenuation of Retinoic Acid (RA) signalling by upregulated WNT in CRC.** In normal colon with wild-type (*wt*) *APC* (**A**), the down-regulation of WNT signaling by *wt-APC* allows signaling through the retinoic acid pathway, which maintains the PNC phenotype of ALDH+ cells. In colon cancer with *APC* mutations (**B**), the upregulation of WNT signaling due to mutant *APC* causes a downregulation of RA signaling, which leads to an increase in the SC phenotype and a decrease in PNC phenotype of ALDH+ cells. Thus, mutant *APC*, via constitutively activated WNT signaling, leads to decreased differentiation of ALDH + SCs along the NEC lineage and an increased number of ALDH+ SCs. Attenuation of retinoic acid signaling provides a mechanism that explains why ALDH+ SCs remain immature in *APC*-mutant colon tissues during CRC tumorigenesis in FAP patients.

example, blocking WNT signaling by the WNT inhibitor Dickkopf-1 was required for the induction of neural markers and neural differentiation of mouse embryonic SCs in response to retinoic acid [134, 141]. The prerequisite that WNT signaling must be downregulated for neural differentiation to be inducible by RA treatment provides a mechanism that helps explain how increased WNT signaling, due to *APC* mutation in CRC development, prevents maturation of ALDH+ colonic SCs along the NEC lineage (Fig 6).

## Summary

One of the properties that SCs have is that are multipotent and possess a mechanism for differentiating. We studied ALDH which is a key component of the RA signaling mechanism that allows SCs to differentiate into NECs. We focused on ALDH and RA signaling with regard to differentiation of NECs, but not other differentiated cell types (e.g. absorptive and goblet

cells), because SCs differentiate into NECs in the SC niche. This study also investigated maturation of SCs along the NEC lineage in FAP tissues because this maturation is altered by the main mechanism that drives CRC development–*APC* mutations lead to SC overpopulation.

The importance of our study comes from two main effects of *APC* mutation in human colonic tissues—SC overpopulation and decreased differentiation of NECs. The importance of the SC overpopulation is readily apparent–more CSCs will produce more malignant daughters which promotes tumor growth. The importance of the decrease in NEC numbers is less apparent. It is well recognized by pathologists [142] that CRCs have reduced numbers of NECs, as was quantified in our study, but other than being a pathological observation, the underlying mechanisms and functional consequences of reduced NECs in human CRCs have not been well understood. In the normal SC niche, NECs are the central communication hub for the crypt. There are several different types of NECs, which constitutes an entire enteric endocrine regulatory system that has different functions and secretes different neuropeptides and hormones that regulates numerous processes both locally in the colon and elsewhere in the body. There are different types of NECs in the colon that communicate via endocrine, paracrine and juxtacrine signals, emphasizing the complexity of the NEC endocrine system. NECs also constitute an enteric nervous system that is regulated by outside neural factors as discussed above. Thus, loss of NECs can have serious functional consequences in malignant colonic tissues.

Given this complexity of the NEC system, it must maintained by an intricate regulatory mechanism with different levels of feedback regulation functioning in the colonic crypt. For example, an important new finding herein is that GLP-2R+ and SSTR1+ PNCs are regulated by feedback mechanisms via their respective GLP-2R+ and SSTR1+ mature NECs, which control different NECs. This regulation is important because GLP-2 and SST neuropeptides produced by NECs have opposing effects on crypt cell proliferation. Moreover, innovative findings from the current study indicate that immature ALDH+ SCs are regulated by crosstalk dynamics between WNT signaling and RA signaling, which likely plays an essential role in maintaining the delicate balance between crypt regeneration and homeostasis in the normal colonic crypt [143, 144]. In CRC tissues that have mutant *APC*, there will be an alteration between WNT and RA signaling whereby WNT signaling is upregulated that leads to attenuation of RA signaling. Thus, a link between WNT and RA signalling provides a plausible mechanism that can explain how mutant *APC* leads to decreased maturation of ALDH+ SCs along the NEC lineage and an increased the number of ALDH+ SCs (Fig 6).

## Conclusion

An important aspect of our research is that is we studied SCs in human colonic crypts using a genetic model (FAP) to investigate how *APC* mutation drives CRC development. This research on human colonic SCs augments the remarkable research that has been done on SCs in mouse small intestinal crypts for many years [reviewed in 76–80, 145]. Moreover, studying the changes that occur in NEC maturation of human colonic SCs due to *APC* mutation could have important clinical relevance. For example, it has been reported [146] that rectal carcinomas treated with radiation or radiation plus chemotherapy have a substantial number of viable tumor NECs that survive after treatment. This finding indicates that malignant NECs are relatively resistant to conventional treatments such as chemotherapy and radiation therapy and these NECs can regenerate tumor growth that leads to cancer recurrences. Indeed, studies on mouse small intestine show that SCs that co-stain for NEC markers can regenerate and revert to a SC state following radiation damage [35, 36]. Based on our research, we surmise that ALDH+/CGA+ PNCs can also regenerate SCs by reverting back to an ALDH+/CGA–SC state.

Thus, discovering ways to target NEC maturation *in vivo* might lead to new more effective treatment strategies for CRC patients.

## Methods

### Acquisition of tissues

In the current study, we investigated normal epithelium from healthy controls and FAP tissues including premalignant (normal-appearing and adenomatous epithelium) and malignant (carcinomas) colonic tissues. Tissues were processed as we previously described [22, 23]. All tissue sections were obtained through the Pathology Department at Thomas Jefferson University. Our study was approved by the IRB of Thomas Jefferson University. Informed written consent was obtained from the subjects (wherever necessary). The patient studies were conducted in accordance with the following ethical guidelines: Declaration of Helsinki, International Ethical Guidelines for Biomedical Research Involving Human Subjects (CIOMS), Belmont Report, and U.S. Common Rule.

### Immunohistochemical mapping

To study the relationship of crypt SCs and NECs in colonic tissues, we used quantitative immunohistochemical mapping of the distribution and size of NE and SC populations along the crypt axis. Immunohistochemical (IHC) and immunofluorescence (IF) methods were done as we previously described [22, 23]. Antibodies used against NEC markers are given in Table 1 (see table footnote). Briefly, formalin-fixed paraffin-embedded 5μm-thick sections of normal and tumorous tissue, obtained from colonic resections of colorectal cancer (CRC) patients. Specimens were deparaffinized using xylene and rehydrated in graded ethanols, followed by epitope demasking by boiled in citrate buffer (Bio Genex, San Ranmon, CA, # HK086-5K) for 15 min in a microwave oven. Endogenous peroxidase blocking with (3%) hydrogen peroxidase buffer and incubation in 10% bovine serum albumin (BSA) was followed by application of primary antibodies diluted in phosphate buffered saline (PBS). The Biotin-streptavidin detection system (DAKO, Philadelphia PA, # K0675) was applied followed by 3,3-diaminobenzidine for 10 min. Counterstaining was performed with hematoxylin and sections were mounted using Fluka DPX mountant (Hauppauge, NY, #06522) and visualized using an Olympus U-SPT microscope. In control experiments, endogenous biotin was blocked by applying avidin for 30 min, followed by biotin for 30 min prior to the application of the biotinylated detection reagent. Negative controls obtained by replacing primary antibody with 10% fetal bovine serum. For immunofluorescence microscopy, deparaffinization was followed by incubation in 10% BSA, application of PBS-diluted 1˚ Ab, fluorophore-attached 2˚ anti-Ab with counter-staining provided by DAPI and mounting using SlowFade Gold antifade reagent (Biotium, Fremont, CA, #S36936). All secondary antibodies were purchased form Molecular Probes anti-human Alexa Fluor 488 (cat # A-11013) and anti-human Alexa Fluor 594 (cat # A-11014). A Zeiss Epi-Fluorescence microscope and software (AxioVision 4.7, Carl Zeiss, Inc.) was used to analyze the tissue sections.

### Indices

Indices (plots of the proportion of cells at each crypt level that stained for any given marker) were constructed from IHC staining patterns as we previously described [23, 110]. Briefly, indexes (proportion of cells at each crypt level that stained for any given marker) were developed from IHC staining patterns. For example, indices for CGA showed the proportion of cells exhibiting this marker at each level along the crypt axis, from which curves showing the

distribution of those cells can be plotted and the total number of CGA$^+$ cells per crypt can be calculated (area under the curve [AUC]). Graphical display of indices and curve fitting (using sixth-order polynomial analysis) were done using Excel (v. 2002, Microsoft, Redmond WA). AUC for plots of these indices was calculated using Prism, a graphical software package from Graph Pad, Inc. (San Diego, CA). The area was used to calculate the proportion of total crypt cells staining positive for the marker and tissue in question. We did not plot any indices for carcinomas because they do not contain recognizable crypt structures. In addition to counting CGA$^+$ and ALDH$^+$ cells as a function of total crypt cells (to develop indexes), we also evaluated, using fluorescence microscopy and double staining to identify cells that were ALDH$^+$/CGA$^+$, ALDH$^+$/CGA$^-$ and ALDH$^-$/CGA$^+$ cells to determine the intracryptal location of each type. For this analysis, only full-length crypts were evaluated (IHC mapping of 24 crypts for each index was done by scoring number of positively stained cells based on crypt level). These data were plotted as scatterplots. We did this for normal, FAP, and adenomatous crypts. In a separate analysis, we calculated the percentage of each type as a fraction of the total number of cells that stained for ALDH$^+$/CGA$^+$ or ALDH$^+$/CGA$^-$ or ALDH$^-$/CGA$^+$. This latter analysis was not limited to full-length crypts. It included carcinomas, which lack crypt structures. These data were plotted as a stacked bar graph.

## Cell culture

Cell lines were obtained from the ATCC (authenticated by cytogenetic analysis) and grown in McCoys 5A medium (HT29, HTB-38; HCT116, CCL-247), L-15 medium (SW480, CCL-228), RPMI-1640 medium (LoVo, CCL-229; COLO320, CCL-220.1) respectively, containing 10% FBS and 1% Penicillin/Streptomycin and cultured at 5% $CO_2$, 37˚C. All experiments were carried out within 10 passages of being thawed. Cells were routinely tested for mycoplasma (Universal mycoplasma detection kit, ATCC Manassas, VA cat# 30-1012K).

## Flow cytometry

Flow cytometry was done as described previously [38, 40]. Briefly, all cells were grown to 70–80% confluency and lifted using an EDTA based solution called Cell Stripper (Fisher Scientific, cat# 25-056-CI). Cells were spun for five minutes to pellet and resuspended in RPMI-1640 with either the mouse monoclonal SSTR1 antibody (Advanced Targeting System, cat # AB-N35) at a 1:100 dilution (10 μg/ml), mouse IgM isotype control at a 10 μg/ml concentration (BioLegend, cat # 401601), mouse monoclonal GLP-2R antibody (R&D Sytems, cat # MAB4285) at a 1:100 dilution (2.5 μg / 200 μl), and goat IgG isotype control at a 2.5 μg / 200 μl concentration (Jackson Laboratories). Cells were incubated on ice for one hour. Following primary antibody and IgG / IgM incubation, cells were washed twice with PBS, and then incubated in the appropriate secondary antibody at a dilution of 1:200 for one hour on ice. All secondary antibodies were purchased form Molecular Probes anti-human Alexa Fluor 488 (cat # A-11013) and anti-human Alexa Fluor 594 (cat # A-11014). Cells were washed twice and resuspended in 500 μl of RPMI-1640 medium and passed through a BD round bottom tube with a 50 56 μm cell strainer (BD Biosciences). Cell surface staining was analyzed using the FACSCalibur and FACSAria II. Flow Cytometry for sorting: All cells were grown to 70–80% confluency and lifted using an EDTA based solution called Cell Stripper (Fisher Scientific, Hampton, NH cat# 25-056-CI). Cells were spun for five minutes to pellet and resuspended in RPMI-1640 with either the mouse monoclonal SSTR1 antibody (Advanced Targeting System, Carlsbad, CA cat # AB-N35) at a 1:100 dilution (10 μg/ml), mouse IgM isotype control at a 10 μg/ml concentration (BioLegend, San Diego, CA cat # 401601), mouse monoclonal GLP-2R antibody (R&D Systems, Minneapolis, MN cat # MAB4285) at a 1:100 dilution (2.5 μg /

200 μl), and goat IgG isotype control at a 2.5 μg / 200 μl concentration (Jackson Laboratories, West Grove, PA).

## ALDEFLUOR assay

The ALDEFLUOR assay was performed according to the manufacturer's protocol (Stem Cell Technologies, Inc., Vancouver, CA cat# 01700) and samples were run on a BD FACS Aria II using the FACSDiva Software as we described [38–41]. Briefly, cells were incubated on ice for one hour. Following primary antibody and IgG / IgM incubation, cells were washed twice with PBS, and then incubated in the appropriate secondary antibody at a dilution of 1:200 for one hour on ice. All secondary antibodies were purchased form Molecular Probes anti-human Alexa Fluor 488 (Invitrogen, Carlsbad, CA cat # A-11013) and anti-human Alexa Fluor 594 (Invitrogen, Carlsbad, CA cat # A-11014). Cells were washed twice and resuspended in 500 μl of RPMI-1640 medium and passed through a BD round bottom tube with a 50 μm cell strainer (BD Biosciences). The BD FACS ARIAII flow cytometer was turned on and given 30 minutes for the laser to warm up. When sorting the colon cancer cell lines, an 85-micron nozzle was used and when sorting primary colon normal and tumor cells, a 100-micron nozzle. Before sorting cells, an initial performance check was done on the machine for all lasers and filters by running the Cytometer setup and Tracking (CST). Once the stream was stable, Accudrop performance was checked to determine the appropriate drop delay setting. For setting up the dot plots, histograms and gates, unstained cells were run on the machine to identify where the P1 main population of cells was on the plot. The cell cluster on the dot plots were adjusted by moving the voltages of the forward scatter (FSC) and side scatter (SSC). Further gating on sequential dot plots allowed for doublet discrimination to ensure single cell populations were analyzed and set up for sorting. Once all these parameters were setup, analysis and sorting was followed using the stained samples. If double-labeled cells were going to be sorted out, compensation controls were ran for each color used in the labeling of cells.

## Isolation and quantification of GLP-2R+ and SSTR1+ cells

GLP-2R+ and SSTR1+ cells were quantified and isolated from cell lines by immunostaining and FACS sorting using anti-GLP-2R (described above) or anti-SSTR1 (described above) antibodies (1:100 dilutions), respectively, and secondary anti-human Alexa-Fluor-488 and anti-human Alexa-Fluor-594 antibodies (both from Molecular Probes, Eugene, OR). In conditioned media (CM) experiments cells were first cultured for 48 hours, media then collected every 24 hours, spun down, and frozen in 1ml aliquots for testing effects of conditioned medium on ALDEFLUOR+ cells.

## Exogenous GLP-2 and SST treatment

The effect of exogenous ligands (GLP-2 Tocris, Minneapolis, MN cat# 2258 and SST Tocris, Minneapolis, MN cat# 1157) on HT29 cells was evaluated as described previously [38]. IC50 dose used for GLP-2 was 500nM.

## Co-culture experiments

Isolated SSTR1+ or GLP-2R+ cells were co-cultured with ALDEFLUOR+ cells for 1 week. Isolated SSTR1− or GLP-2R− cells co-cultured with ALDEFLUOR+ cells were used as controls. ALDEFLUOR+, SSTR1+, SSTR1−, GLP-2R+, and GLP-2R− cell subpopulations were isolated by FACS from the HT29 cell line. The number of cells was counted using a Countess Cell counter (Invitrogen, Carlsbad, CA).

### Effect of SST on SSTR1+ cells & ALDEFLUOR+ cells, and on expression of the MCM2 proliferative cell marker

The effect of SST on cell subpopulations in colonospheres was tested using the colonosphere assay as previously described [38–41]. Briefly, ALDEFLUOR+ cells isolated from the HT29 line were grown in presence of SST or SST inhibitor (cycloSST, Tocris, Minneapolis, MN cat# 3493) under ultra-low attachment conditions for 10 days. Colonospheres were trypsinized to dissociate single cells. ALDH+ cells were enumerated using the ALDEFLUOR assay and SSTR1 cells quantified by immunostaining and FACS using anti-SSTR1 antibody (described above). The effect of SST on expression of the MCM2 proliferative cell marker in colonospheres formed from ALDEFLOUR+ cells in presence of SST or cycloSST was determined by western blotting analysis of pooled experimental samples (n = 6) and results were quantified by densitometry. Western blotting was performed as previously described [38–41] using MCM2 antibody (1:100; Abcam, Cambridge, UK cat# ab108935).

### Mathematical modeling

We constructed a mathematical model (see S1 File) that was based on the biologic data from normal and neoplastic colonic epithelium. Our model was created based on the three compartment model of Nakata et al [147] in order to simulate the 3 different cell populations: SCs, PNCs, and NECs. The model incorporates a feedback loop whereby the proportion of NECs regulates the rate of SC division and the rate of PC division. This assumption was based on the fact that many feedback mechanisms in growth processes in biology involve feed back stemming from the products to the reactants in order to regulate the rate of generation of the product. Thus, we assumed that the mature cells feed back either on stem cells or on progenitor cells.

We first did iterative fitting to find model parameters that correspond to proportions of SCs, PCs and NECs in normal, FAP, adenoma and CRC (see S1 Table in S1 File). Once we obtained a set of values for model parameters that fit well with the biological data, we determined how the probability of SC self-renewal, the rate of SC division, the death rate for NECs, and the SC reproduction number changed in modeling neoplastic tissue vs. normal colon.

### Statistical analysis

An Unpaired t-test was used to determine significance between the treatment group and the appropriate controls. All the values obtained with a p-value less than 0.05 were considered to be statistically significant.

### Supporting information

**S1 File.**
(PDF)

### Acknowledgments

We thank Dr Olaf Runquist at Hamline University for his support, the Thomas Jefferson University Pathology Core Facility personnel for their assistance, Deni Galileo for his technical advice, Joseph George for his participation, and Tim Flanagan for assistance in graphics services with Figure preparation. A special thanks to Drs Goldstein and Isenberg in the Division of Colorectal Surgery, Thomas Jefferson University for providing tissue specimens for this

study. Finally, the authors would like to thank Dr. Nicholas Petrelli for his support for the *in vitro* experiments that were conducted at the Helen F. Graham Cancer Center.

## Author Contributions

**Conceptualization:** Brooks Emerick, Shirin R. Modarai, Gilberto Schleiniger, Bruce M. Boman.

**Data curation:** Tao Zhang, Shirin R. Modarai, Lynn M. Opdenaker, Gilberto Schleiniger, Bruce M. Boman.

**Formal analysis:** Tao Zhang, Koree Ahn, Brooks Emerick, Shirin R. Modarai, Gilberto Schleiniger, Jeremy Z. Fields, Bruce M. Boman.

**Funding acquisition:** Shirin R. Modarai, Gilberto Schleiniger, Bruce M. Boman.

**Investigation:** Tao Zhang, Koree Ahn, Brooks Emerick, Shirin R. Modarai, Lynn M. Opdenaker, Juan Palazzo, Gilberto Schleiniger, Bruce M. Boman.

**Methodology:** Tao Zhang, Koree Ahn, Brooks Emerick, Shirin R. Modarai, Lynn M. Opdenaker, Gilberto Schleiniger, Bruce M. Boman.

**Project administration:** Gilberto Schleiniger, Jeremy Z. Fields, Bruce M. Boman.

**Resources:** Lynn M. Opdenaker, Juan Palazzo, Bruce M. Boman.

**Software:** Brooks Emerick, Gilberto Schleiniger.

**Supervision:** Lynn M. Opdenaker, Juan Palazzo, Jeremy Z. Fields, Bruce M. Boman.

**Validation:** Shirin R. Modarai.

**Visualization:** Tao Zhang, Koree Ahn, Brooks Emerick, Shirin R. Modarai, Lynn M. Opdenaker, Juan Palazzo, Jeremy Z. Fields, Bruce M. Boman.

**Writing – original draft:** Brooks Emerick, Jeremy Z. Fields, Bruce M. Boman.

**Writing – review & editing:** Jeremy Z. Fields, Bruce M. Boman.

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
