## [Decision Letter · Decision Letter 0]

22 Jun 2020

PONE-D-20-14837

APC mutations in human colon lead to decreased maturation of ALDH+ stem cells along the neuroendocrine lineage: Discovery of a link between WNT and retinoic acid signalling in colon cancer development

PLOS ONE

Dear Dr. Boman,

Thank you for submitting your manuscript to PLOS ONE. After careful consideration, we feel that it has merit but does not fully meet PLOS ONE’s publication criteria as it currently stands. Therefore, we invite you to submit a revised version of the manuscript that addresses the points raised during the review process.

Please modify the manuscript according to the reviewers' suggestions

We look forward to receiving your revised manuscript.

Kind regards,

Irina V. Lebedeva, Ph.D.

Academic Editor

PLOS ONE

Journal Requirements:

2. In your Methods section, please provide additional details regarding the tissues used in your study. Please include the source from which you obtained the tissue, and the catalog number or internal reference number if applicable. For more information on PLOS ONE's guidelines for research using cell lines, see https://journals.plos.org/plosone/s/submission-guidelines#loc-cell-lines.

3. Our journal requires that methods are described in enough detail to allow suitably skilled investigators to fully replicate your study. If materials, methods, and protocols are well established, authors may cite articles where those protocols are described in detail, but the submission should include sufficient information to be understood independent of these references. Please revise your manuscript so that cited protocols are briefly but sufficiently described. For more information please see https://journals.plos.org/plosone/s/submission-guidelines#loc-materials-and-methods.

4. In your methods section, please include the sources, catalog numbers, and dilutions of all antibodies used in your experiments. In addition, we note that you state "CGA antibody was from Dako and used at a dilution of 1:5." Please confirm that this is the correct dilution.

5. In your methods section, please include the catalog numbers of all cell lines used in your experiments.

6.Thank you for stating the following in the Competing Interests section:

'The authors have declared that no competing interests exist.'

We note that one or more of the authors are employed by a commercial company: 5CATX,

Inc.

7. We note that you have included the phrase “data not shown” in your manuscript. Unfortunately, this does not meet our data sharing requirements. PLOS does not permit references to inaccessible data. We require that authors provide all relevant data within the paper, Supporting Information files, or in an acceptable, public repository. Please add a citation to support this phrase or upload the data that corresponds with these findings to a stable repository (such as Figshare or Dryad) and provide and URLs, DOIs, or accession numbers that may be used to access these data. Or, if the data are not a core part of the research being presented in your study, we ask that you remove the phrase that refers to these data.

9. We note that Figure 1 in your submission contain copyrighted images. All PLOS content is published under  the Creative Commons Attribution License (CC BY 4.0), which means that the manuscript, images, and Supporting Information files will be freely available online, and any third party is permitted to access, download, copy, distribute, and use these materials in any way, even commercially, with proper attribution. For more information, see our copyright guidelines: http://journals.plos.org/plosone/s/licenses-and-copyright.

Additional Editor Comments (if provided):

Reviewers' comments:

Reviewer's Responses to Questions

**Comments to the Author**

1. Is the manuscript technically sound, and do the data support the conclusions?

Reviewer #1: Yes

Reviewer #2: Partly

2. Has the statistical analysis been performed appropriately and rigorously? 

Reviewer #1: No

Reviewer #2: I Don't Know

3. Have the authors made all data underlying the findings in their manuscript fully available?

Reviewer #1: Yes

Reviewer #2: Yes

4. Is the manuscript presented in an intelligible fashion and written in standard English?

Reviewer #1: Yes

Reviewer #2: Yes

5. Review Comments to the Author

Reviewer #1: In the manuscript by Zhang et al, the authors investigate the relationship between APC mutations and stem cell maturation in colon cancer. They use patient samples, mathematical modeling, and in vitro studies to show that APC mutation delays stem cell maturation and to connect Wnt and retinoic acid (RA) signaling. The principles of the current manuscript are important and add to the scientific understanding of colon cancer progression; however, concerns outlined below dampen the strength of these findings.

Major concerns:

1. The authors state, in the abstract and introduction, that they are investigating a correlation between Wnt signaling and RA signaling. The data shown in this manuscript have not proven this interaction.

2. The staining for ALDH1 and CGA in Figure 1 are described as having a similar pattern (1A and B – normal). However, the images shown do not depict this.

3. The authors state that “the proportion of SSTR+ cells tended to be greater than that of GLP-2R+ cells” in the colon cancer cell lines. The data shown in Figure 5 depict that one of the cell lines (COLO320) has some differences between SSTR1 and GLP-2R, but the cell lines chosen for future studies (SW480 and HT29) show no difference in the percentage of positive cells for SSTR1 and GLP-2R. The in vitro studies are lacking error and significance designations (Figure 5B-H), making the data difficult to interpret. In addition, controls are lacking in Figure 5B. The authors state that GLP-2 decreases the ALDH+ cell population, but that SST does not. Without showing the control cells, it is impossible to discern the effect of SST and GLP-2.

4. The results describe data of CGA/ALDH double staining in FAP tissues, but FAP staining isn’t shown (Figure 2). It would be helpful to show the IF staining for normal, FAP, adenoma, and carcinoma, since those are shown in panel “F.” In addition, it would be helpful to show the crypt level for all four conditions.

Minor concerns:

1. Layout of the results is difficult to follow. It would be helpful if the results and figures were in the same order.

2. In general, the figures are of poor quality. In addition, the IHC and IF images are lacking scale bars and would benefit from arrows to depict areas of interest. The text in most figures needs to be improved, and this is particularly evident in Figure 4E and F, where the green text is illegible. Similarly, the panels in Figure 5D don’t show anything.

3. It would be helpful to have the crypt levels shown on one example IHC and IF image to help orient the reader.

4. The y-axes in Figure 2 need to be labeled. As it is now, C-E show the crypt level and then there is a switch to % of the cell population. This is not labeled and left up to reader interpretation.

5. In the results and figure legend, Figure 3 is said to have multiple specific panels; however, the image only shows A and B (B is never described).

Reviewer #2: This paper by Zhang et al. describes the relationship between APC mutations, ALDH+ stem cells (SCs) and maturation along the neuroendocrine lineage in colonic crypts from FAP patients. The authors major conclusion is that progression from normal colon to adenomatous colon in APC mutant FAP patients is driven by a failure of ALDH+ SCs to differentiate into mature neuroendocrine cells (NECs). Immature intermediate progenitor NECs self-renew and proliferate causing overpopulation of the SC niche, leading to adenoma formation. The authors use IHC and IF on colon tissue from FAP tissue to make observations of NEC and SC marker positivity. Confirmatory and mechanistic experiments are performed in vitro using colon cancer cell lines. Mechanistically, the authors conclude that APC-mutant driven WNT signaling drives retinoic acid (RA) signaling to prevent maturation of colonic SCs into NECs. Overall, the studies are well conducted. However, some major and minor deficiencies prevent the paper from being suitable for publication in its current form.

MAJOR:

1. The studies are largely observational and descriptive in nature. The title indicates a link between RA and WNT signaling, but this is not mechanistically tested in this paper. Instead, the authors cite a number of previously studies to support their conclusions.

2. The mechanistic studies focus on GLP-2 and SST signaling, which are well conducted and support the major conclusions of the paper. The title should mention these mechanistic observations.

3. Many of the conclusions are based on amalgamation of previously published work. The conclusions should be tempered and based on data presented in the current paper.

4. The model in Figure 4 refers to a feedback mechanism involving the regulation of progenitor SC proliferation by mature NECs. It’s not clear that this model is fully supported by the data presented.

MINOR:

1. Page 5: Reference to 1/2 or 1/20 individuals who develop adenomatous polyps or colon cancer, respectively. Do these numbers refer to the general population? Please clarify.

2. Page 7: 5-HT is referred to as a neuro-peptide. This is incorrect. It’s a catecholamine.

3. Figure 1C, D and E: curve axes should have the same scale to enable more meaningful comparison.

4. Figure 2B: ALDH staining is much weaker in colon cancer specimens compared to normal crypts. What’s the explanation?

5. Quantitation of data in Figure 3 should be shown, similar to Figure 2.

6. Figure 3 panels are mislabeled as A and B. The text refers to panels A through J.

7. Raw flow cytometry data should be included for plots in Figure 5. The raw data could be added as supplemental material.

6. PLOS authors have the option to publish the peer review history of their article (what does this mean?). If published, this will include your full peer review and any attached files.

Reviewer #1: No

Reviewer #2: No

---

## [Author Response · Author response to Decision Letter 0]

18 Aug 2020

We thank the reviewers for their helpful comments. Each reviewer comment (italics) is followed by our response (►).

Reviewer #1: 

Major concerns:

1. The authors state, in the abstract and introduction, that they are investigating a correlation between Wnt signaling and RA signaling. The data shown in this manuscript have not proven this interaction.

►We have toned down our main conclusion so that it doesn’t imply that we have proven a correlation between WNT signaling and RA signaling, but that our study provides a clue or hypothesis that a link exists between WNT signaling and RA signaling. This link is the mechanism that explains our results on FAP.

2. The staining for ALDH1 and CGA in Figure 1 are described as having a similar pattern (1A and B – normal). However, the images shown do not depict this.

►We apologize. We have now revised the wording in the text that refers to Figure 1 to indicate that results based on the staining indices show that the crypt distribution of CGA+ cells and ALDH+ cells are similar.

3. The authors state that “the proportion of SSTR+ cells tended to be greater than that of GLP-2R+ cells” in the colon cancer cell lines. The data shown in Figure 5 depict that one of the cell lines (COLO320) has some differences between SSTR1 and GLP-2R, but the cell lines chosen for future studies (SW480 and HT29) show no difference in the percentage of positive cells for SSTR1 and GLP-2R. The in vitro studies are lacking error and significance designations (Figure 5B-H), making the data difficult to interpret. In addition, controls are lacking in Figure 5B. The authors state that GLP-2 decreases the ALDH+ cell population, but that SST does not. Without showing the control cells, it is impossible to discern the effect of SST and GLP-2.

►We thank the reviewer for pointing this out. The sentence stating “the proportion of SSTR+ cells tended to be greater than that of GLP-2R+ cells” is now deleted. We have now added error and significance designations for in vitro studies in Figures 5A-5D to make the data easier to interpret. In addition, we have now added controls in Figures 5B-5D to make it possible to discern the effect of SST, GLP-2, and conditioned medium. Note that the figure panels have been reassigned different letters based on incorporating these revisions.

4. The results describe data of CGA/ALDH double staining in FAP tissues, but FAP staining isn’t shown (Figure 2). It would be helpful to show the IF staining for normal, FAP, adenoma, and carcinoma, since those are shown in panel “F.” In addition, it would be helpful to show the crypt level for all four conditions. 

►The purpose of the graphic in Figures 2A & 2B was to show that ALDH+ cells co-stain with many CGA+ cells in normal crypts, but ALDH+ cells typically do not co-stain with CGA+ cells in CRCs. The other purpose of the data given in Figures 2C-2F was to show our quantification of the crypt location and proportion of cells staining positively for CGA and ALDH1 in FAP tissues. This study involved a large analysis to score stained cells in hundreds of co-stained crypt sections and was not designed to record images to qualitatively show the distribution of stained cells in individual crypts that was already shown in Figures 1A and 1B. Nonetheless, we now provide representative images of co-stained crypts in the supplemental materials section. We also now show the crypt level for all figures in the main text.

Minor concerns:

1. Layout of the results is difficult to follow. It would be helpful if the results and figures were in the same order.

►We have revised the order of the results to make them easier to follow in parallel with the figures.

2. In general, the figures are of poor quality. In addition, the IHC and IF images are lacking scale bars and would benefit from arrows to depict areas of interest. The text in most figures needs to be improved, and this is particularly evident in Figure 4E and F, where the green text is illegible. Similarly, the panels in Figure 5D don’t show anything.

►Thank you for this suggestion. We have now added scale bars and arrows pointing to areas of interest in IHC and IF images. The text in the figures has been improved including Figure 4E and F. We have now removed the panels in Figure 5D but still kept the quantification of ALDH+ cells from this co-culture experiment which is given in Figure 5E. The PDF images of all figures will of course be changed to higher quality TIFF images when the paper is published. 

3. It would be helpful to have the crypt levels shown on one example IHC and IF image to help orient the reader.

►Thank you. We have now added crypt levels to IHC and IF images.

4. The y-axes in Figure 2 need to be labeled. As it is now, C-E show the crypt level and then there is a switch to % of the cell population. This is not labeled and left up to reader interpretation.

►The y-axes in Figure 2 are now labeled in our revised manuscript.

5. In the results and figure legend, Figure 3 is said to have multiple specific panels; however, the image only shows A and B (B is never described).

►We apologize and have corrected this error.

Reviewer #2: 

MAJOR:

1. The studies are largely observational and descriptive in nature. The title indicates a link between RA and WNT signaling, but this is not mechanistically tested in this paper. Instead, the authors cite a number of previous studies to support their conclusions.

►We have toned down our main conclusion so that it doesn’t imply that we have proven a correlation between WNT signaling and RA signaling, but that our study provides a clue or hypothesis that a link exists between WNT signaling and RA signaling. This link is the mechanism that explains our results on FAP.

2. The mechanistic studies focus on GLP-2 and SST signaling, which are well conducted and support the major conclusions of the paper. The title should mention these mechanistic observations.

►The title has been changed to reflect mechanistic observations involving delay in neuroendocrine cell maturation and altered GLP-2 and SST feedback signaling.

3. Many of the conclusions are based on amalgamation of previously published work. The conclusions should be tempered and based on data presented in the current paper.

►We have toned down our main conclusion to indicate that the data in the current study (in context with our previous published data) provides a clue/hypothesis to a link between RA and WNT signaling in CRC development. This idea was conceived during the current study in considering a mechanism that explains our research findings. Since this mechanism has not been described before, we believe that it is an innovative aspect of our study and worthwhile stating particularly since our results provide a strong clue to the possibility for the mechanism.

4. The model in Figure 4 refers to a feedback mechanism involving the regulation of progenitor SC proliferation by mature NECs. It’s not clear that this model is fully supported by the data presented.

►Our assumption was that cell division is regulated by a feedback mechanism. This assumption was based on the fact that many feedback mechanisms in growth processes in biology involve feed back stemming from the products to the reactants in order to regulate the rate of generation of the product. Thus, we assumed that the mature cells feed back either on stem cells or on progenitor cells. In our iterative modeling to fit the data, the model fit with feedback occurring between mature cells and progenitor cells, not between mature cells and stem cells. Indeed, when we performed the biological experiments, we did not find any feedback between mature cells and stem cells only between mature cells and progenitor cells. Thus, the biological data concurred with the modeling result. We have revised the text to clarify the rationale for the assumption of a feedback mechanism in the model and what model results predict regarding feedback signaling.

MINOR:

1. Page 5: Reference to 1/2 or 1/20 individuals who develop adenomatous polyps or colon cancer, respectively. Do these numbers refer to the general population? Please clarify.

►They refer to the general population. We have clarified this point and including citation of the reference.

2. Page 7: 5-HT is referred to as a neuro-peptide. This is incorrect. It’s a catecholamine.

►We apologize and have corrected this mistake.

3. Figure 1C, D and E: curve axes should have the same scale to enable more meaningful comparison.

►We have now added scale bars to Figures 1C, 1D and 1E to enable more meaningful comparison.

4. Figure 2B: ALDH staining is much weaker in colon cancer specimens compared to normal crypts. What’s the explanation?

►The difference in ALDH staining intensity in colon cancer specimens compared to normal crypts is due to technical variation. We have now added a different image of ALDH staining in normal crypts that is a more appropriate comparison with that of colon cancer.

5. Quantitation of data in Figure 3 should be shown, similar to Figure 2.

►The crypt location of the neuroendocrine cell types in Figure 3 is provided in Table 1. We now mention in the text that the crypt distribution of these other NEC types (noted in Table 1) is similar to that of CGA+ cells (Figure 1C). Because the proportion of these other NEC types was small, we were not able to create scatter plots similar to those in Figure 2. The main purpose of Figure 3 was to show that staining for other NEC markers co-stains with CGA, and that SSTR1 cells and GLP-2R cells are different NEC types which each express their own respective neuropeptides SST and GLP.

6. Figure 3 panels are mislabeled as A and B. The text refers to panels A through J.

►We apologize and have corrected this mistake.

7. Raw flow cytometry data should be included for plots in Figure 5. The raw data could be added as supplemental material.

►We have now included this data in supplemental material.

---

## [Decision Letter · Decision Letter 1]

10 Sep 2020

APC mutations in human colon lead to decreased neuroendocrine maturation of ALDH+ stem cells that alters GLP-2 and SST feedback signaling: Clue to a link between WNT and retinoic acid signaling in colon cancer development

PONE-D-20-14837R1

Dear Dr. Boman,

We’re pleased to inform you that your manuscript has been judged scientifically suitable for publication and will be formally accepted for publication once it meets all outstanding technical requirements.

Within one week, you’ll receive an e-mail detailing the required amendments. **Please also address the minor issues highlighted by Reviewer #1**. When these have been addressed, you’ll receive a formal acceptance letter and your manuscript will be scheduled for publication.

Kind regards,

Irina V. Lebedeva, Ph.D.

Academic Editor

PLOS ONE

Additional Editor Comments (optional):

Reviewers' comments:

Reviewer's Responses to Questions

**Comments to the Author**

1. If the authors have adequately addressed your comments raised in a previous round of review and you feel that this manuscript is now acceptable for publication, you may indicate that here to bypass the “Comments to the Author” section, enter your conflict of interest statement in the “Confidential to Editor” section, and submit your "Accept" recommendation.

Reviewer #1: All comments have been addressed

Reviewer #2: All comments have been addressed

2. Is the manuscript technically sound, and do the data support the conclusions?

Reviewer #1: Yes

Reviewer #2: Yes

3. Has the statistical analysis been performed appropriately and rigorously? 

Reviewer #1: Yes

Reviewer #2: Yes

4. Have the authors made all data underlying the findings in their manuscript fully available?

Reviewer #1: Yes

Reviewer #2: Yes

5. Is the manuscript presented in an intelligible fashion and written in standard English?

Reviewer #1: Yes

Reviewer #2: Yes

6. Review Comments to the Author

Reviewer #1: The authors have done a nice job addressing previous comments and concerns.

In the revised Figure 5, it is unclear which cells are being used in the experiments for panels D-H. In addition, while the authors cite the findings here, there are not errors and statistical information on panels E-H.

Reviewer #2: The authors have adequately addressed the major and minor concerns. The paper is suitable for publication.

7. PLOS authors have the option to publish the peer review history of their article (what does this mean?). If published, this will include your full peer review and any attached files.

Reviewer #1: No

Reviewer #2: No

---

## [Editor Report · Acceptance letter]

30 Sep 2020

PONE-D-20-14837R1 

*APC* mutations in human colon lead to decreased neuroendocrine maturation of ALDH+ stem cells that alters GLP-2 and SST feedback signaling: Clue to a link between WNT and retinoic acid signalling in colon cancer development 

Dear Dr. Boman:

I'm pleased to inform you that your manuscript has been deemed suitable for publication in PLOS ONE. Congratulations! Your manuscript is now with our production department. 

Kind regards, 

on behalf of

Dr. Irina V. Lebedeva 

Academic Editor

PLOS ONE